# INFINITEHIP: EXTENDING LANGUAGE MODEL CONTEXT UP TO 3 MILLION TOKENS ON A SINGLE GPU

## ABSTRACT

In modern large language models (LLMs), handling very long context lengths presents significant challenges as it causes slower inference speeds and increased memory costs. Additionally, most existing pre-trained LLMs fail to generalize beyond their original training sequence lengths. To enable efficient and practical long-context utilization, we introduce *InfiniteHiP*, a novel and practical LLM inference framework that accelerates processing by dynamically eliminating irrelevant context tokens through a modular hierarchical token pruning algorithm. Our method also allows generalization to longer sequences by selectively applying various RoPE adjustment methods according to the internal attention patterns within LLMs. Furthermore, we offload the key-value cache to host memory during inference, significantly reducing GPU memory pressure. As a result, InfiniteHiP enables the processing of up to 3 million tokens on a single L40s 48GB GPU – 3x larger – without any permanent loss of context information. Our framework achieves an 18.95x speedup in attention decoding for a 1 million token context without requiring additional training. We implement our method in the SGLang framework and demonstrate its effectiveness and practicality through extensive evaluations.

## 1 INTRODUCTION

In modern Transformer-based generative large language models (LLMs), extending the context length is essential for improving comprehension and coherence in long-context, multi-modal, and retrieval-augmented language generation. However, achieving this poses significant challenges, primarily due to the attention mechanism (Vaswani et al., 2017), a fundamental component of these models. The attention mechanism computes relationships between each input token and all preceding tokens, causing computational and memory costs to scale quadratically with sequence length increases. Another problem arising from the attention mechanism is the key-value (KV) cache. During generation, previously computed attention keys and values are cached in GPU memory for reuse. However, the KV cache size scales linearly, challenging long context inference.

Various methods have been proposed to reduce the high costs of the attention mechanism. FlashAttention (FA2) (Dao et al., 2022) significantly reduces memory and bandwidth use by avoiding writing the entire attention score matrix to global GPU memory. However, it does not reduce the arithmetic cost. Other approaches (Xiao et al., 2024b; Lee et al., 2024b) selectively attend to a fixed number of key tokens, either statically or dynamically, during attention inference.

Many efforts have also been made to mitigate the memory burden of the KV cache. KV cache eviction methods selectively 'forget' past contexts to conserve GPU memory (Zhang et al., 2023; Oren et al., 2024). However, these methods permanently erase past contexts, which may be needed again later. HiP attention (Lee et al., 2024b) offloads infrequently accessed 'cold' tokens to larger and cheaper host memory, dynamically fetching them back to GPU during generation only when needed while keeping only frequently accessed 'hot' tokens on the GPU.

Despite these optimizations, another problem with context extension still remains: pre-trained LLMs cannot handle inputs longer than their trained context length. Since the attention mechanism is permutation invariant, they utilize positional embedding methods such as Rotary Positional Embeddings (RoPE) Su et al. (2023) to model the temporal order of tokens. However, as LLMs are

Figure 1: **Overview of InfiniteHiP.** *(a) Infinitely growing KV cache:* In InfiniteHiP, the context keys and values are stored in a unified memory space, where some of the keys and values are loaded on GPU memory. *(b) Configurable modular pruning:* Each pruning stage narrows down the candidate key indices based on the current query block. During pruning, if a cache miss is encountered, the missing tokens are dynamically loaded and the GPU cache is updated. *(c) Paged block sparse attention:* The selected key indices are used to perform efficient paged block sparse attention.

typically pre-trained on sequences truncated to a fixed length, they fail to adapt to unseen positions when prompted with longer contexts.

One option for overcoming this problem is long context fine-tuning (Rozière et al., 2024), i.e., fine-tuning the model on a set of longer inputs. However, fine-tuning, especially on long sequences, requires exorbitant training costs and high-quality training data. Thus, *out-of-length* (OOL) generalization, i.e., the capability for pre-trained models to perform well beyond their pre-trained limits without training, becomes increasingly important. Self-Extend (Jin et al., 2024) proposes a training-free way of scaling the RoPE embeddings beyond the pre-trained limit.

In this paper, we propose *InfiniteHiP*, a long-context LLM framework that combines the strengths of all the above methods. To alleviate the computational burden of attention, InfiniteHiP proposes a novel modular sparse attention scheme that minimizes computation for less important contexts. For optimizing KV cache offloading, InfiniteHiP enhances HiP attention (Lee et al., 2024b)'s offloading strategy with a sophisticated LRU-based cache policy. Finally, InfiniteHiP achieves OOL generalization by carefully applying various RoPE adjustment strategies within different components of LLMs according to their internal attention patterns. By providing a unified solution to all the aforementioned problems as a whole, InfiniteHiP demonstrates strong practicality and is well suited for real-world deployment.

What sets InfiniteHiP apart is its novel use of pruning modules, as illustrated in Figure 1. These modules employ a novel modular hierarchical pruning algorithm to selectively discard less important input tokens. The algorithm leverages common patterns observed in attention matrices of popular LLMs – namely, their sparsity and the spatial locality of nonzero entries within a sequence – to prune irrelevant tokens effectively. Each pruning module partitions the input sequence into chunks of fixed length $b_k$, and efficiently identifies the approximate top-1 token with the highest attention score within each chunk in parallel. Only the top-$K$ most significant chunks (where $K$ is constant) are passed to the next module, while the rest are discarded. By stacking multiple pruning modules, InfiniteHiP iteratively refines a block sparse attention mask.

While our work is based upon HiP (Lee et al., 2024b), we overhaul several key mechanisms. First, our novel hierarchical pruning modules achieve higher accuracy compared to HiP's heuristic-based hierarchical pruning. Second, the pruning algorithm within each module is significantly faster due to its enhanced parallelizability. Lastly, its modular design enables fine-grained control over pruning-stage caches, leading to much faster decoding than HiP.

InfiniteHiP enables extremely long-context inference with pre-trained LLMs, surpassing their original context length limits without quality degradation while overcoming GPU memory limitations with efficient KV cache offloading. As a training-free solution, InfiniteHiP can be used as a drop-in replacement for any pretrained Transformer-based LLM, providing faster inference and extending usable context length at both the model and hardware levels.

Our contributions can be summarized as follows:

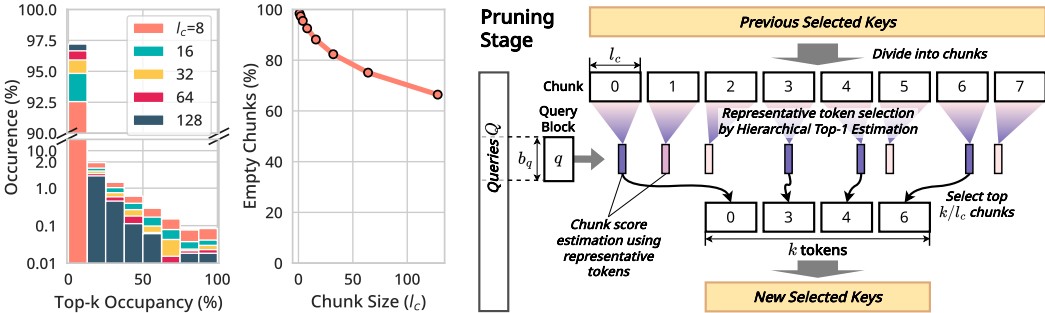

(a) **Chunk sparsity.** In the given 128K context, *Left:* A histogram which plots the frequency of chunks ($y$) which contain a certain percentage ($x$) of the top 2048 keys. *Right:* Percentage of chunks that contain none of the top 2048 keys by varying chunk size ($l_c$). We use the Llama 3.1 8B model and extract data from one of the attention layers.

(b) **Modular context pruning.** We design our context pruning module based on the observation in (a). A single pruning stage is shown above. The keys selected in the previous stage are divided into chunks, and a representative token is selected for each chunk. Each chunk's score is estimated from these representative tokens. Finally, the top $l_c/k$ chunks are selected for the next stage.

Figure 2: **Design of our Context Pruning Algorithm.**

- We propose a modular, highly parallelizable training-free hierarchically pruned attention mechanism that enables out-of-length generalization while significantly speeding up LLM inference on long contexts.

- We demonstrate that our method does not degrade the LLM's long-context language understanding and reasoning capabilities compared to other SoTA efficient long-context inference methods.

- We efficiently implement InfiniteHiP on the SGLang LLM serving framework, achieving a 7.24× speedup in end-to-end decoding on a 3M token context while using only 3.34% of the VRAM required by FA2, and design an efficient KV cache offloading algorithm that utilizes modular pruning algorithm, making it practical for real-world scenarios.

## 2    RELATED WORKS

Previous studies have proposed dynamic token selection for efficient LLM inference for long contexts. MInference (Jiang et al., 2024) classifies attention heads into two types to estimate the sparse attention pattern, which is used to drop less important tokens before the dot product. While this method considerably speeds up the prefill stage, it cannot be applied in the decoding stage, which takes up most of the inference time. HiP Attention (Lee et al., 2024b) estimates the top-$k$ context blocks with the highest attention scores in a hierarchical and iterative manner, significantly speeding up both prefill and decoding in long contexts. However, the iterative algorithm involves many global thread synchronizations, which hinders parallelism. Quest (Tang et al., 2024) divides the context into fixed-size pages and estimates the maximum attention score by using cached element-wise min and max vectors. InfLLM (Xiao et al., 2024a) divides the context sequence into blocks and selects representative tokens in each block. For each new query, the top-$k$ blocks whose representative tokens give the highest attention scores are selected. In contrast to our InfiniteHiP, the representative tokens of each block are prechosen and do not change with the current query. Both HiP and InfLLM enable KV cache offloading, which makes long context inference possible within a single GPU.

## 3    FROM HIP TO INFINITEHIP

We point out three major problems of the previous HiP (Lee et al., 2024b) and our proposed changes to address those problems in InfiniteHiP approach as follows:

**Problem 1: Low Parallelizability of Hierarchical Top-$k$ Estimation**

**Problem**    In HiP (Lee et al., 2024b)'s Hierarchical Top-$k$ Estimation, each iteration contains a global top-$k$ operation which selects $k$ chunks out of the $2k$ candidates, which limits the degree of parallelism per attention head to $k$ threads (typically set to 1024), regardless of the context length. Furthermore, each iteration requires a global synchronization, so $O(\log_2 T)$ global synchronizations are needed (where $T$ is the number of context tokens).

**Solution**    InfiniteHiP overhauls the token pruning algorithm to allow a higher degree of parallelism and require fewer global thread synchronizations. This is done by splitting the context sequence into

$O(T)$ chunks of fixed size, instead of $O(1)$ chunks of variable size as in HiP. This is motivated by the chunk sparsity of attention scores, which suggests that the top-$k$ tokens are concentrated in few contiguous context chunks, shown in Figure 2a.

Also, just a few (3 in our default setting) global thread synchronizations are required at each of our novel pruning stages. While this change increases the time complexity of the pruning algorithm from HiP's $O(\log T)$ to $O(T)$, the increased parallelizability means that InfiniteHiP's pruning algorithm runs faster on modern GPUs in practice. See Section 3.1 for an in-depth description of our token pruning algorithm.

## Problem 2: No Out-of-length Generalization Capability

**Problem** Speeding up long-context inference with 3 million tokens is not useful if existing pre-trained models' generation quality drops significantly after a 32K token threshold. HiP is capable of doing the former, but its usefulness is severely limited by pre-trained models that do not support out-of-length generalization.

**Solution** InfiniteHiP employs a dynamic RoPE adjustment trick to allow OOL generalization of any pre-trained short-context model. See Section 3.2 for more details.

## Problem 3: Inefficient KV Cache Offloading

**Problem** While HiP proposes a preliminary method to offload the KV cache to the host memory to reduce pressure on the GPU VRAM. However, it incurs a large overhead during top-$k$ estimation process, because which elements will be accessed from the host memory is inherently unpredictable.

**Solution** InfiniteHiP addresses this problem by caching each pruning stage's output candidates, and refreshing each of them at different intervals. The first pruning stage, which is the most costly, is refreshed least frequently, and each subsequent stages are refreshed more often. This strikes a balance between performance and accuracy. Furthermore, we use the Least Recently Used (LRU) policy for efficient GPU cache management. More details are described in Section 3.3.

Figure 3: **InfiniteHiP (Ours) vs. HiP performance. Left:** Accuracy vs. context length. **Center:** LongBench results. **Right:** Single-layer attention latency (ms).

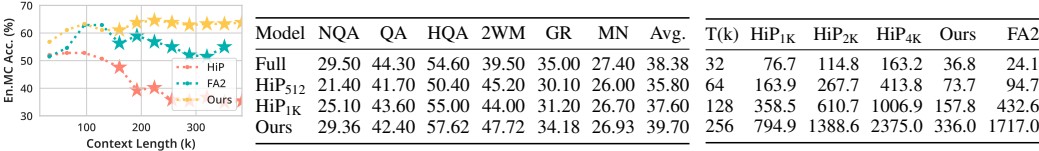

| Model | NQA | QA | HQA | 2WM | GR | MN | Avg. | T(k) | HiP$_{1K}$ | HiP$_{2K}$ | HiP$_{4K}$ | Ours | FA2 |
|---|---|---|---|---|---|---|---|---|---|---|---|---|---|
| Full | 29.50 | 44.30 | 54.60 | 39.50 | 35.00 | 27.40 | 38.38 | 32 | 76.7 | 114.8 | 163.2 | 36.8 | 24.1 |
| HiP$_{512}$ | 21.40 | 41.70 | 50.40 | 45.20 | 30.10 | 26.00 | 35.80 | 64 | 163.9 | 267.7 | 413.8 | 73.7 | 94.7 |
| HiP$_{1K}$ | 25.10 | 43.60 | 55.00 | 44.00 | 31.20 | 26.70 | 37.60 | 128 | 358.5 | 610.7 | 1006.9 | 157.8 | 432.6 |
| Ours | 29.36 | 42.40 | 57.62 | 47.72 | 34.18 | 26.93 | 39.70 | 256 | 794.9 | 1388.6 | 2375.0 | 336.0 | 1717.0 |

As a result of our solution, we could achieve superior performance and efficiency in long-context than the previous HiP, as shown in Fig. 3.

### 3.1 Efficient Multi-Stage Context Pruning

In this section, we introduce a novel and efficient design for context pruning. A complete description of our algorithm is detailed in Appendix A. Here, we describe the key points of our design.

**Background.** Given query, key, and value sequences $\boldsymbol{Q}, \boldsymbol{K}, \boldsymbol{V} \in \mathbb{R}^{H \times T \times d}$, the conventional multi-head attention output $\boldsymbol{O}$ is computed as $\boldsymbol{O} = \text{Concat}[\boldsymbol{O}_1, \ldots, \boldsymbol{O}_H]$, where $\boldsymbol{S}_h = \boldsymbol{Q}_h \boldsymbol{K}_h^\top \in \mathbb{R}^{T \times T}$, $\boldsymbol{P}_h = \text{softmax}(\boldsymbol{S}_h) \in \mathbb{R}^{T \times T}$, $\boldsymbol{O}_h = \boldsymbol{P}_h \boldsymbol{V}_h \in \mathbb{R}^{T \times d}$ for all $h = 1..H$, where $H$ denotes the number of attention heads, $T$ denotes the sequence length, $d$ denotes the embedding dimension, and softmax is applied row-wise Vaswani et al. (2017). The causal masking and constant scaling are omitted for brevity. The $\boldsymbol{S}$ and $\boldsymbol{P}$ matrices are each called the *attention scores* and *probabilities*.

Note that the initial $n_{\text{sink}}$ tokens (*sink* tokens) and $n_{\text{stream}}$ most recent tokens (*streaming* tokens) are always included. We sparsely select the tokens in between the sink and streaming tokens. We aim to find a block sparse attention mask that approximately selects the top-$K$ key blocks with the highest attention scores for each query block. This allows us to perform efficient block sparse attention (BSA) while preserving the capabilities of the model (Lee et al., 2024b). For conciseness, in this section, we ignore the existence of sink and streaming tokens, as well as the causal part of the self-attention.

**Overview of our Method.** Unlike HiP, we use multiple pruning stages to find the top-$k$ tokens, each discarding context chunks irrelevant to the current query. By applying the pruning stages, InfiniteHiP generates a sparse attention mask, a good approximation for the top-$k$ tokens.

Figure 2b illustrates how each pruning stage preserves only the most relevant contexts. First, the input key tokens are partitioned into equally sized chunks of fixed size (in contrast to HiP, which divides the tokens into a fixed number of chunks). Next, we select a representative token for each key chunk. In HiP, the middle token was always chosen for every chunk. In contrast, InfiniteHiP chooses the representative token dynamically: we use a top-1 variant of the Hierarchical Mask Selection Algorithm (HMSA) (Lee et al., 2024b). Note that our use of HMSA is to find the representative token for each chunk, not by itself for selecting the top-k, which contrasts our method to HiP's. Additionally, since this HMSA is performed locally within each chunk, there is no need for global GPU thread synchronizations. See Appendix A.1.3 for details on our modification of HMSA.

Using the attention scores of these representative tokens, max-pooled across attention heads, we select the top-$K$ key chunks and discard the rest. The surviving tokens are used as the input for the next pruning stage. By iteratively applying these pruning stages, we can effectively obtain a good estimate of the top-$k$ tokens in the form of a sparse attention mask.

**Efficient Modular Context Pruning.** Formally, we denote a pruning stage by $\mathcal{S}^{(i)} = (b_q^{(i)}, l_c^{(i)}, k^{(i)})$, where $b_q$ is the size of the query block, $l_c$ is the chunk size, $k$ is the number of tokens to keep, and the superscript $i = 1 \ .. \ N$ denotes the stage index. To enable parallel processing, the queries are grouped into contiguous blocks. Specifically, in the $i$th stage, the query $\boldsymbol{Q}$ is divided into multiple $b_q^{(i)}$-sized blocks. $\boldsymbol{q}_{h,m}^{(i)} \in \mathbb{R}^{b_q \times d}$ denotes the $h$th attention head's $m$th query block in the $i$th pruning stage.

For the initial stage, we select all of the key indices $\mathcal{I}_m^{(0)} = [1, \ldots, T]$ for each query block index $m$. Each pruning stage will transform this list of indices into a smaller list by discarding indices corresponding to less important contexts.

In each $i$th stage, the input sequence $\mathcal{I}_m^{(i-1)}$ is divided into contiguous chunks of size $l_c^{(i)}$ where the $j$th chunk contains $\mathcal{C}_{m,j}^{(i)}$. From each $\mathcal{C}_{m,j}^{(i)}$, we dynamically pick a representative token independently for each head, using a top-1 variant of the algorithm used in HiP (See Appendix A.1.3 for details). The representative token index for the $h$th head is denoted by $r_{h,m,j}^{(i)} = \text{SelectRep}(\boldsymbol{q}_{h,m}^{(i)}, \mathcal{C}_{m,j}^{(i)})$.

The representative tokens provide a way to estimate the maximum attention score within each chunk. We estimate each chunk's score by computing the maximum value across the heads and each query in the query block as $s_{m,j}^{(i)} := \max_{\substack{h=1..H, \\ t=1..b_q^{(i)}}} (\boldsymbol{q}_{h,m}^{(i)})_t^\top \boldsymbol{k}_{h, r_{h,m,j}^{(i)}}$. Finally, the top $K^{(i)} := k^{(i)}/l_c^{(i)}$ chunks with the highest estimated attention scores are selected for the next stage, as follows:

$$\mathcal{I}_{m'}^{(i)} = \bigcup_{\hat{j} \in \mathcal{T}_m^{(i)}} \mathcal{C}_{m,\hat{j}}^{(i)}, \text{ where } \mathcal{T}_m^{(i)} = \arg\text{top}_{K^{(i)}}(s_{m,j}^{(i)}), \text{ and } m' = \begin{cases} \lceil m \cdot b_q^{(i)}/b_q^{(i+1)} \rceil & \text{if } i \le N, \\ m & \text{otherwise.} \end{cases} \quad (1)$$

When all $N$ stages are done, we are left with sparse key indices $\mathcal{I}_m^{(N)} \in \{1, \ldots, T\}^{k^{(N)}}$ for all query blocks $m = 1 \ .. \ T/b_q^{(N)}$, which can be used for efficient block sparse attention, also used in existing sparse attention methods (Lee et al., 2024b; Jiang et al., 2024; Lai et al., 2025).

## 3.2 DYNAMIC RoPE FOR OOL GENERALIZATION

We employ a novel combination of multiple RoPE interpolation strategies for the sparse key tokens for out-of-length generalization. During token pruning, two strategies are employed: (1) **Chunk-indexed RoPE:** Each key chunk is given a single position ID, where the last chunk's position ID is offset by $n_\text{stream}$ from the current query. All keys in the chunk are given the same position ID. (2) **Relative-style RoPE:** During the hierarchical top-1 estimation algorithm, the left branch gets a position ID offset by $n_\text{stream} + 1$ from the current query, and the right branch gets a position ID offset by $n_\text{stream}$ from the current query. For chunk score estimation, the representative key is given a position ID offset by $n_\text{stream}$ from the current query. We apply strategy (1) for the first three layers of the LLM and strategy (2) for the rest. The reason for this choice is explained in detail in Appendix D. During block sparse attention, we use the StreamingLLM-style RoPE: The selected keys, including the sink and streaming keys, are given position IDs sequentially in their original order, where the most recent token is given

| Method | Window | Synthetic Tasks | | | | | NLU | | | | Avg. Abs. | Avg. Rel.(%) |
|---|---|---|---|---|---|---|---|---|---|---|---|---|
| | | RPK | RN | RKV | MF | Avg. | MC | QA | SUM | Avg. | | |
| FA2 | 8K | 8.50 | 7.80 | 6.20 | 21.70 | 11.05 | 44.10 | 15.50 | 24.70 | 28.10 | 19.57 | 47.83 |
| NTK | 128K | 0.00 | 0.00 | 0.00 | 2.60 | 0.65 | 0.00 | 0.40 | 6.40 | 2.27 | 1.46 | 3.65 |
| SelfExtend | 128K | **100** | **100** | 0.20 | 22.60 | 55.70 | 19.70 | 8.60 | 14.70 | 14.33 | 35.02 | 67.81 |
| Infinite | 8K | 6.80 | 7.60 | 0.20 | 20.60 | 8.80 | 41.50 | 14.60 | 20.80 | 25.63 | 17.22 | 42.52 |
| Streaming | 8K | 8.50 | 8.30 | 0.40 | 21.40 | 9.65 | 40.60 | 14.30 | 20.40 | 25.10 | 17.38 | 42.53 |
| H2O | 8K | 2.50 | 2.40 | 0.00 | 6.00 | 2.73 | 0.00 | 0.70 | 2.80 | 1.17 | 1.95 | 3.95 |
| InfLLM | 8K | **100** | 99.00 | 5.00 | **23.70** | 56.92 | 43.70 | 19.50 | 24.30 | 29.17 | 43.05 | 89.07 |
| **InfiniteHiP** | **3K** | 99.83 | 97.46 | 9.60 | 17.71 | 56.15 | 57.21 | 26.94 | 24.89 | 36.35 | 46.25 | 98.17 |
| **InfiniteHiP** | **5K** | **100** | 99.83 | **10.80** | 20.00 | **57.66** | 55.90 | **30.99** | 22.63 | 36.50 | **47.08** | **99.69** |

Table 1: ∞**Bench Results on Llama 3 8B.** The average score of each category is the mean of the dataset performance, and the average score of the whole benchmark is the relative performance compared to the best-performing result. See Table 22 for more results.

Table 2: **LongBench.** See Table 1 for notations.

| | | Avg. Abs. | Avg. Rel.(%) |
|---|---|---|---|
| Methods | Window | Llama 3 8B | |
| FA2 | 8K | 42.47 | 87.69 |
| Infinite | 8K | 40.62 | 83.23 |
| Streaming | 8K | 40.61 | 83.21 |
| InfLLM | 8K | 44.47 | 92.83 |
| **InfiniteHiP** | **3K** | 47.72 | 100.00 |

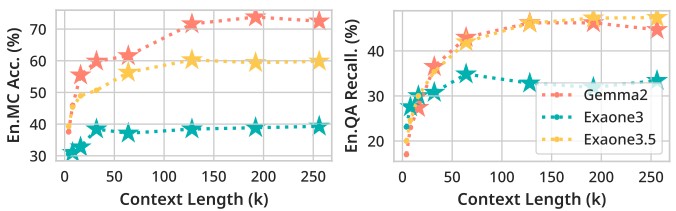

Figure 4: **Results with Short Context Models**. Star (⋆)-shaped markers indicate out-of-length generalization results.

the same position ID as the current query (Xiao et al., 2024b). Since this dynamic RoPE incurs some computational overhead, it can be disabled when the OOL generalization is unnecessary.

### 3.3 KV CACHE OFFLOADING

We improve the KV cache offloading mechanism of HiP Attention (Lee et al., 2024b) by enhancing its cache management policy. Similarly to HiP Attention, we manage the KV cache on the unified memory space while keeping a smaller key bank on the GPU memory, which acts as a cache. Note that we maintain two different key banks on the GPU for the mask-selection and block sparse-attention processes. We also keep a page table, which maps the global key index to an index within the GPU key bank, in the GPU memory as well. Upon a cache miss, the missing keys are fetched from the unified memory space and placed on the GPU bank. Unlike HiP Attention (Lee et al., 2024b), we use the Least Recently Used (LRU) policy as the eviction mechanism.

**Sparse Attention Mask Caching.** To further reduce latency during decoding, we cache the sparse attention mask for each pruning stage. We observe that the sparse attention mask exhibits temporal locality. Therefore, instead of recomputing it every decoding step, we update the output attention mask for the $i$th pruning stage periodically every $n_{\text{refresh}}^{(i)}$ steps using the latest query block. Additional details are provided in Appendix A.

**Implementation.** We implement the GPU kernels for our method using the Triton language (Tillet et al., 2019). We implement a single GPU kernel for the pruning stage, which can be reused for all stages just with different parameters. For block sparse attention, we implement a method similar to FlashAttention (Dao et al., 2022) for prefill and Flash Decoding (Dao et al., 2023) for decoding. We also combine PagedAttention (Kwon et al., 2023) to alleviate the overhead from KV cache memory management. To implement dynamic loading and offloading with host memory, we use Nvidia UVM (Unified Virtual Memory).

## 4 EXPERIMENTS

### 4.1 EXPERIMENT SETTING

**Hyperparameters and Baselines.** We describe hyperparameter details in Appendix F. We compare the performance of InfiniteHiP against the following baselines, mostly chosen for their long-context

| | T=256k | | T=512k | | T=1024k | |
|---|---|---|---|---|---|---|
| | VRAM (GB) | Latency (μs) | VRAM (GB) | Latency (μs) | VRAM (GB) | Latency (μs) |
| FA2[(*)] (1M window) | 20.0 (100%) | 1,193 (100%) | 36.0 (100%) | 2,325 (100%) | 68.0 (100%) | 4,645 (100%) |
| InfLLM (12K) | 4.8 (23.8%) | 1,186 (99.4%) | 4.8 (13.2%) | 1,194 (51.4%) | 4.8 (6.99%) | 1,234 (26.6%) |
| Ours (Fast) | 6.1 (30.4%) | 532 (44.6%) | 6.1 (16.9%) | 902 (38.8%) | 6.1 (8.93%) | 1,864 (40.1%) |
| Ours (Flash) | 6.1 (30.4%) | 325 (27.2%) | 6.1 (16.9%) | 475 (20.4%) | 6.1 (8.93%) | 844 (18.2%) |

Table 3: **Decoding Attention Latency of InfiniteHiP with Offloading.** The latencies are measured with a single RTX 4090 on PCIe 4.0 x8. The model used is AWQ Llama3.1 with FP8 KV cache. (*) FA2 does not support KV cache offloading and thus cannot run decoding with a context window exceeding 128K tokens using a single RTX 4090. We estimate FA2 results by layer-wise simulation with the same model architecture.

| | Prefill (ms) | | | | | | Decode (us) | | | | | |
|---|---|---|---|---|---|---|---|---|---|---|---|---|
| T (k) | 128 | 256 | 384 | 512 | 768 | 1024 | 128 | 256 | 384 | 512 | 768 | 1024 |
| FA2 (1M window) | 379 | 821 | 1267 | 1711 | 2602 | 3490 | 643 | 1193 | 1787 | 2325 | 3457 | 4645 |
| InfLLM (12K) | 178 | 179 | 180 | 181 | 182 | 183 | 1157 | 1174 | 1167 | 1182 | 1203 | 1222 |
| HiP (1K) | 109 | 122 | 135 | 135 | 147 | 147 | 376 | 399 | 423 | 423 | 446 | 450 |
| **Ours** (3K) | 84.5 | 96.7 | 109 | 122 | 147 | 172 | 89.5 | 103 | 124 | 154 | 195 | 234 |
| **Ours** + Extend (3K) | 138 | 158 | 178 | 197 | 236 | 276 | 98.0 | 111 | 133 | 163 | 205 | 245 |

Table 4: **Attention Latency Comparison of InfiniteHiP and baselines.** Prefill latency is measured with chunked prefill, with a chunk size of 32K. Ours uses the 3K preset from Table 1.

capabilities. (1) **Truncated FA2**: The input context is truncated in the middle to fit in each model's pre-trained limit, and we perform dense attention with FlashAttention2 (FA2) (Dao et al., 2022). (2) **DynamicNTK** (bloc97, 2023) and (3) **Self-Extend** (Jin et al., 2024) adjust the RoPE for OOL generalization. We perform dense attention with FA2 without truncating the input context for these baselines. Both (4) **LM-Infinite** (Han et al., 2024) and (5) **StreamingLLM** (Xiao et al., 2024b) use a combination of sink and streaming tokens while also adjusting the RoPE for OOL generalization. (6) **H2O** (Zhang et al., 2023) is a KV cache eviction strategy which retains the top-$k$ KV tokens at each decoding step. (7) **InfLLM** (Xiao et al., 2024a) selects a set of representative tokens for each chunk of the context, and uses them for top-$k$ context selection. (8) **HiP Attention** (Lee et al., 2024b) uses a hierarchical top-$k$ token selection algorithm based on attention locality.

**Benchmarks.** We evaluate the performance of InfiniteHiP on mainstream long-context benchmarks. (1) LongBench (Bai et al., 2023), whose sequence length averages at around 32K tokens, and (2) ∞Bench (Zhang et al., 2024) with a sequence length of over 100K tokens. Both benchmarks feature a diverse range of tasks, such as long document QA, summarization, multi-shot learning, and information retrieval. We apply our method to the instruction-tuned models. As our framework is training-free, applying our method to these models has zero extra cost.

## 4.2 RESULTS

**LongBench.** In Table 2, our method achieves about 7.17%p better relative score using Llama 3 and 3.19%p better using Mistral 0.2 compared to the best-performing baseline, InfLLM. This is significant because our method processes 4× fewer key tokens through sparse

Table 5: **RULER Performance Comparison by Input Length.**

| Method | 4k | 8k | 16k | 32k | 64k | 128k | **Avg.** |
|---|---|---|---|---|---|---|---|
| Full | 96.74 | 94.03 | 92.02 | 84.17 | 81.32 | 76.89 | 87.52 |
| FlexPrefill | 95.99 | 93.67 | 92.73 | 88.14 | 81.14 | 74.67 | 87.72 |
| MInference | 96.54 | 94.06 | 91.37 | 85.79 | 83.03 | 54.12 | 84.15 |
| SeerAttn | 84.43 | 79.55 | 79.80 | 72.95 | 64.79 | 51.61 | 72.18 |
| Xattn S=8 | 96.83 | 94.07 | 93.17 | 90.75 | 84.08 | 72.31 | 88.47 |
| Ours | 96.24 | 94.43 | 93.66 | 88.17 | 84.38 | 74.99 | **88.64** |

attention compared to InfLLM, leading to better decoding latency as shown in Table 4.

**RULER.** In Table 5, we compare our method with baselines (Jiang et al., 2024; Xu et al., 2025; Lai et al., 2025; Gao et al., 2025) on RULER (Hsieh et al., 2024). Baseline measurements are measured by Xu et al. (2025). In this experiment, we use dense decode exceptionally, due to the lack of baselines' sparse decode support. Additionally, dense decoding appears necessary to recover the original model's performance in this benchmark; a detailed discussion is provided in Appendix E.2.

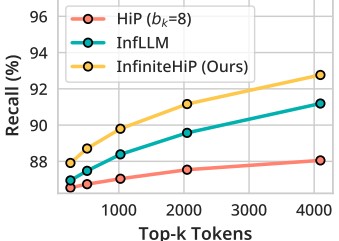

Figure 6: **Top-$k$ Recall.**

| | |
|---|---|
| FA2 (128K) | 67.25 |
| Ours ($N = 2$) | 70.31 |
| Ours ($N = 3$) | **74.24** |

Table 7: **Ablation on No. of Stages in ∞Bench En-MC.**

Table 8: **RoPE Ablation Study in Context Pruning and Sparse Attention.** We measure the accuracy of ∞Bench En.MC subset truncated with $T$=128K with various combinations of RoPE extends style in context pruning and sparse attention kernels. Each row represents a single RoPE extend style in the context pruning procedure, and each column represents the RoPE extend style in block sparse attention.

| RoPE Style in Pruning \ SA | DE | IL | ST | AVG. |
|---|---|---|---|---|
| DE (Dynamic) | 52.40 | 54.59 | 51.09 | 52.69 |
| IL (InfLLM) | 68.12 | 66.81 | **70.31** | 68.41 |
| CI (Chunk-Indexed) | 67.69 | 66.81 | 67.69 | 67.39 |
| RT (Relative) | 66.81 | 68.56 | **70.31** | **68.56** |
| AVG. | 63.76 | 64.19 | **64.85** | - |

**∞Bench.** We show our results on ∞Bench in Table 1. Our *3K-fast* and *3K-flash* options use the same setting as *3K* except they use longer mask refreshing intervals as detailed in Section 4.1. Our method achieves 9.99%p better relative score using Llama 3 and 4.32%p better using Mistral 0.2 compared to InfLLM. The performance gain is larger than in LongBench, which has a fourfold shorter context. This suggests that our method is able to better utilize longer contexts than the baselines.

To further demonstrate our method's superior OOL generalization ability, we compare ∞Bench's En.MC score in various context lengths with Llama 3.1 8B in Fig. 3 (left). While InfiniteHiP keeps gaining performance as the context length gets longer, baselines with no OOL generalization capability degrade significantly beyond the pretrained context length (128K). In Fig. 4, we experiment with other short-context LLMs: Exaone 3 (4K) (LG AI, 2024a), Exaone 3.5 (32K) (LG AI, 2024b) and Gemma2 (8K) (Gemma Team, 2024). We observe the most performance gain in an extended context with these short-context models. For instance, with Gemma2, we gain an impressive +24.45%p in En.MC and +22.03%p in En.QA compared to FA2.

**LongVideoBench.** We show our performance with multi-modality on Llama 4 Scout (AI, 2025) and Qwen2.5 VL 32B (Qwen et al., 2025) on LongVideoBench (Wu et al., 2024). We could recover most of the full dense attention performance with only 1.54% degradation in average.

Table 6: **LongVideoBench Result.**

| | T (k) | FA | Ours |
|---|---|---|---|
| Llama4 Scout 109B | 256 | 52.27 | 51.07 |
| Qwen2.5 VL 72B | 128 | 56.15 | 54.28 |

### 4.3 ANALYSIS

In this section, we analyze the latency and the effect of each of the components of our method.

**Latency.** We analyze the latency of our method on a 1-million-token context and compare it against baselines with settings that yield similar benchmark scores. In Table 4, we measure the latencies of attention methods. During a 1M token prefill, our method is

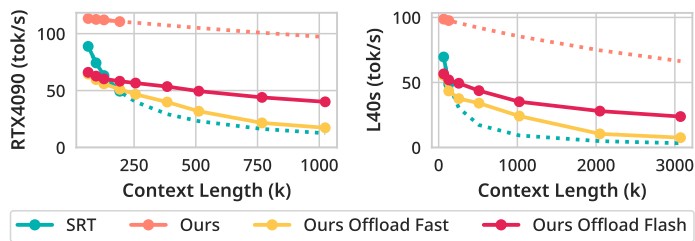

Figure 5: **SGlang Decoding Throughput Benchmark.** Dashed lines are estimated values. RTX 4090 has 24GB and L40s has 48GB of VRAM. We use AWQ Llama3.1 with FP8 KV cache.

20.29× faster than FlashAttention2 (FA2), 6% faster than InfLLM, and achieves similar latency with the baseline HiP. During decoding with a 1M token context, our method significantly outperforms FA2 by 19.85×, InfLLM by 4.98×, and HiP by 92%. With context extension (dynamic RoPE) enabled, our method slows down about 1.6× in prefill and 5% in decoding due to overheads incurred by additional memory reads of precomputed cos and sin vectors. Therefore, our method is 50% slower than InfLLM in context extension-enabled prefill, but it is significantly faster in decoding because decoding is memory-bound: Our method with a 3K token context window reads fewer context tokens than InfLLM with a 12K token context window.

Table 9: **Ablations study of context extrapolation method on InfiniteBench ($T$=128K).**

| Llama 3.1 8B | MC | QA-Recall | QA-F1 | Avg. | Qwen3 8B | Training Free | MC | QA-Recall | QA-F1 | Avg. |
|---|---|---|---|---|---|---|---|---|---|---|
| Full Model | 63.75 | 44.82 | 36.11 | 48.23 | YaRN Only | ✗ | 59.38 | 33.78 | 17.10 | 36.75 |
| Ours - RoPE adj. | 61.57 | 36.28 | 30.60 | 42.82 | YaRN + Ours | ✗ | 60.26 | 35.57 | 19.20 | 38.34 |
| Ours + RoPE adj. | 67.24 | 48.28 | 34.10 | 49.87 | Vanilla + Ours | ✓ | 59.82 | 37.01 | 23.65 | 40.16 |

**Latency with KV Offloading.** In Table 3, we measure the decoding latency with KV cache offloading enabled on a Passkey retrieval task sample. We keep FA2 in the table for reference, even though FA2 with UVM offloading is 472× slower than the baseline HiP. Among the baseline methods, only InfLLM achieves KV cache offloading in a practical way. In 256K context decoding, we outperform InfLLM by 3.64×. With KV cache offloading, the attention mechanism is extremely memory-bound, because accessing the CPU memory over PCIe is 31.5× more expensive in terms of latency than accessing VRAM. InfLLM chooses not to access the CPU memory while executing its attention kernel, so it has to sacrifice the precision of its top-k estimation algorithm. This makes larger block and context window sizes necessary to maintain the model's performance on downstream tasks. In contrast, we choose to access the CPU memory during attention kernel execution like baseline HiP. This allows more flexibility for the algorithm design, performing better in downstream NLU tasks. Moreover, our UVM implementation makes the KV cache offloaded attention mechanism a graph-capturable operation, which allows us to avoid CPU overheads, unlike InfLLM. In contrast to the offloading framework proposed by Lee et al. (2024b), we cache the sparse attention mask separately for each pruning stage. This enables us to reduce the frequency of calling the costly initial pruning stage, which scales linearly.

**Throughput.** In Fig. 5, we present the decoding throughput of our method using RTX 4090 (24GB) and L40S (48GB) GPUs. On the 4090, our method achieves a throughput of 3.20× higher at a 1M context length compared to the estimated decoding throughput of SRT (SGlang Runtime with FlashInfer). Similarly, on the L40S, our method surpasses SRT by 7.25× at a 3M context length. Due to hardware limitations, we estimated the decoding performance since a 1M and 3M context requires approximately 64GB and 192GB of KV cache, respectively, which exceeds the memory capacities of 24GB and 48GB GPUs. We further demonstrate that adjusting the mask refreshing interval significantly enhances decoding throughput without substantially affecting performance. The *Flash* configuration improves decoding throughput by approximately 3.14× in a 3M context compared to the *Fast* configuration.

**Accuracy of top-$k$ estimation.** In Fig. 6, we demonstrate our method has better coverage of important tokens, which means higher recall of attention probabilities of selected key tokens. Our method performs 1.57%p better than InfLLM and 4.72%p better than baseline HiP. The better recall indicates our method follows pretrained attention patterns more closely than the baselines.

**Ablation on Depth of Stage Modules.** In Table 7, we perform an ablation study on a number of stages ($N$) that are used in ours. The latency-performance optimal pruning module combination for each setting is found empirically.

**Ablation on RoPE interpolation strategies.** In Table Table 9, we show context extrapolation method effects (left), and compare with the training required method YaRN (right). We show that our context extrapolation method can even improve the performance of the model with minimal latency overhead. And also our method outperforms YaRN (Peng et al., 2023), in a standalone manner, despite our method being able to cooperate with YaRN. In Table 8, we perform an ablation study on the dynamic RoPE extrapolation strategy in masking and sparse attention. We choose the best-performing RT/ST combination for our method.

## 5 CONCLUSION

In this paper, we introduced *InfiniteHiP*, a training-free LLM inference framework for efficient long context inference that supports out-of-length generalization and dynamic KV cache offloading. InfiniteHiP effectively addresses the three major challenges that arise in long context LLM inference: (1) Efficient inference with long contexts, (2) Out-of-length generalization, (3) GPU memory conservation through KV cache offloading without 'forgetting'. The experiments on LongBench and ∞Bench, and the latency benchmarks demonstrate our method's superior performance and practicality over previous state-of-the-art methods.

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

## A    COMPLETE DESCRIPTION OF ALGORITHMS

### A.1    CONTEXT PRUNING

We describe our multi-stage context pruning algorithm in Algorithm 1, which uses the pruning stage described in Algorithm 2.

#### A.1.1    MULTI-STAGE CONTEXT PRUNING.

Our pruning algorithm generates a sparse binary mask $M$ of size $T_q/b_q \times T_{kv}$ for each attention layer, where $T_q$ is the length of the queries, $b_q$ is the size of each query block, and $T_{kv}$ is the length of the keys. This sparse binary mask can be more efficiently represented in memory with a set of arrays of indices $\{\mathcal{I}_m\}_{m=1}^{T_q/b_q}$, where $\mathcal{I}_m$ contains every integer $j$ such that $M_{m,j} \neq 0$.

---

**Algorithm 1** InfiniteHiP Context Pruning Algorithm

---

**input** Number of pruning stages $N$, Pruning stages $\mathcal{S}^{(1)}, \dots, \mathcal{S}^{(N)}$, where each stage $\mathcal{S}^{(i)} = (b_q^{(i)}, l_c^{(i)}, k^{(i)})$, Query length $T_q$, Key length $T_{kv}$, Number of sink tokens $n_{\text{sink}}$, Number of streaming tokens $n_{\text{stream}}$.

1:  $\mathcal{I}_m^{(0)} := [n_{\text{sink}}, \dots, b_q^{(1)} \cdot m - n_{\text{stream}}]$ for $m = 1 \mathinner{\ldotp\ldotp} T_q/b_q^{(1)}$.  ▷ *Exclude sink and streaming tokens*
      *without breaking causality.*
2:  **for each** pruning stage $i = 1 \mathinner{\ldotp\ldotp} N$ **do**
3:      **for each** query block $m = 1 \mathinner{\ldotp\ldotp} T_q/b_q^{(i)}$ **do**
4:          $\mathcal{I}_m'^{(i)} := \text{PruningStage}(\mathcal{S}^{(i)}, \mathcal{I}_m^{(i-1)})$ **if** not cached. (Algorithm 2)
5:          **for all** $m'$ such that $m' = \lceil m \cdot b_q^{(i)}/b_q^{(i+1)} \rceil$ **do**
6:              $\mathcal{I}_{m'}^{(i)} := \mathcal{I}_m'^{(i)}$.  ▷ *Subdivide query blocks for the next stage.*
7:          **end for**
8:      **end for**
9:  **end for**
10: **return** resulting mask indices $\mathcal{I}_m^{(N)}$ for $m = 1 \mathinner{\ldotp\ldotp} T_q/b_q^{(N)}$.

---

#### A.1.2    PRUNING STAGE.

Each pruning stage narrows down the selection of key tokens for a given query block.

---

**Algorithm 2** InfiniteHiP Pruning Stage (PruningStage)

---

**input** Pruning stage $\mathcal{S} = (b_q, l_c, k)$, Previous stage's key indices $\mathcal{I}_m$ for the $m$th query block, Queries $Q \in \mathbb{R}^{H \times T_q \times d}$, Keys $K \in \mathbb{R}^{H \times T_{kv} \times d}$, where $H$ is the number of attention heads, $T_q$ and $T_{kv}$ are the number of query and key tokens each, and $d$ is the model dimension, Current layer index $l$.

**output** Filtered key indices $\mathcal{I}_m'$.

1:  $n_{\text{block}} := T_q/b_q$.
2:  $q_{h,m} := Q_{h, m \cdot b_q : (m+1)b_q - 1}$ for $h = 1 \mathinner{\ldotp\ldotp} H$.  ▷ *Divide the queries into $n_{block}$ blocks for each head.*
3:  $\tilde{q}_{h,m} := \text{ApplyRopeQ}_l(q_{h,m})$.
4:  $n_{\text{chunk}} := |\mathcal{I}_m|/l_c$.
5:  $\mathcal{C}_j := [\mathcal{I}_m[j\, l_c], \dots, \mathcal{I}_m[(j+1)l_c - 1]]$ for $j = 1 \mathinner{\ldotp\ldotp} n_{\text{chunk}}$.  ▷ *Divide the key indices into $n_{chunk}$*
      *chunks.*
6:  **for each** chunk $j = 1 \mathinner{\ldotp\ldotp} n_{\text{chunk}}$ **do**
7:      **for each** head $h = 1 \mathinner{\ldotp\ldotp} H$ **do**
8:          $r_{h,m,j} := \text{SelectRep}(q_{h,m}, \mathcal{C}_j)$. (Algorithm 3)  ▷ *Select the representative token for this chunk.*
9:      **end for**
10:     $\tilde{k}_{h, r_{h,m,j}} := \text{ApplyRopeK}_{l,2}(k_{h, r_{h,m,j}})$.
11:     $s_{m,j} := \max_{h=1..H, t=1..b_q} [\tilde{q}_{h,m}]_t^\top \tilde{k}_{h, r_{h,m,j}}$.  ▷ *Compute the estimated chunk attention score.*
12: **end for**
13: $\mathcal{T}_m := \arg \text{top}_{k/l_c}(s_{m,j})$.  ▷ *Discard chunks with low estimated attention scores.*
      $\phantom{x}_j$
14: $\mathcal{I}_m' := \bigcup_{\hat{j} \in \mathcal{T}} \mathcal{C}_{\hat{j}}$.

---

### A.1.3 REPRESENTATIVE TOKEN SELECTION USING HMSA.

Although largely unchanged from the hierarchical mask selection algorithm first introduced in Lee et al. (2024b), we present it again in Algorithm 3 for completeness. The difference from the original HiP is in what the algorithm is used for; in HiP, it was used directly to select the sparse attention mask, but in our InfiniteHiP, it is used to select the representative token in each chunk.

In short, a key chunk is divided into two equal-sized subchunks (branches), and the score of the first token of each branch is evaluated to select the more 'important' branch. The branch with the lower score is discarded, and the remaining branch is subdivided again to select the more 'important' branch. This process is repeated until the sub-branch's length is one token wide and we find the representative token.

The SelectRep algorithm is designed to approximately estimate the the location of the top-1 key token with the highest attention score in the given key chunk, without evaluating all of the keys in the chunk. It runs in $O(\log_2 l_c)$ time, where $l_c$ is the key chunk size.

---

**Algorithm 3** Representative Token Selection (SelectRep) by Hierarchical Top-1 Selection (Lee et al., 2024b)

---

**input** Query block $\boldsymbol{q} \in \mathbb{R}^{b_q \times d}$, Key chunk indices $\mathcal{C} \in \mathbb{N}^{l_c}$, Keys $\boldsymbol{K} \in \mathbb{R}^{T_{kv} \times d}$, Layer index $l$.
**output** A representative token index $r \in \mathcal{C}$.
1: $\tilde{\boldsymbol{q}} := \text{ApplyRopeQ}_l(\boldsymbol{q})$.
2: $\boldsymbol{k} := \begin{bmatrix} \boldsymbol{K}_{\mathcal{C}_1} & \cdots & \boldsymbol{K}_{\mathcal{C}_{l_c}} \end{bmatrix}^\top \in \mathbb{R}^{l_c \times d}$.     ▷ *Load key tokens with the given indices.*
3: $n_{\text{iter}} := \lceil \log_2(l_c) \rceil$.
4: $(n_{\text{first}}^{(1)}, n_{\text{last}}^{(1)}) := (1, l_c)$.
5: **for each** iteration $i = 1 .. n_{\text{iter}}$ **do**
6:   $m^{(i)} := \lfloor (n_{\text{first}}^{(i)} + n_{\text{last}}^{(i)})/2 \rfloor$.
7:   $\left( \mathcal{B}_1^{(i)}, \mathcal{B}_2^{(i)} \right) := \left( (n_{\text{first}}^{(i)} : m^{(i)} - 1), (m^{(i)} : n_{\text{last}}^{(i)}) \right)$.
8:   **for each** branch index $j = 1 .. 2$ **do**
9:     Pick the first index $r_j^{(i)}$ from the range $\mathcal{B}_j^{(i)}$.
10:     $\tilde{\boldsymbol{k}} \leftarrow \text{ApplyRopeK}_{l,j}(\boldsymbol{k}_{r_j^{(i)}})$.
11:     Compute scores $\sigma_j^{(i)} := \max_t \left( \tilde{\boldsymbol{q}}_t^\top \tilde{\boldsymbol{k}} \right)$.
12:   **end for**
13:   $t^{(i)} := \arg\max_j \sigma_j^{(i)}$.              ▷ *Pick the top-1 index.*
14:   $(n_{\text{first}}^{(i+1)} : n_{\text{last}}^{(i+1)}) := \mathcal{B}_{t^{(i)}}^{(i)}$.           ▷ *Update range.*
15: **end for**
16: $r := n_{\text{first}}^{(n_{\text{iter}})}$.

---

The ApplyRopeQ and ApplyRopeK functions used in Algorithms 2 and 3 are defined as follows.

$$\text{ApplyRopeQ}_l(\boldsymbol{q}) := \begin{cases} \text{ApplyRope}(\boldsymbol{q}, \boldsymbol{p}[n_{\text{stream}} + 1]) & \text{if } l > 3 \\ \text{ApplyRope}(\boldsymbol{q}, \boldsymbol{p}[\min\{i_{\text{orig}}, l_c + n_{\text{stream}}\}]) & \text{otherwise,} \end{cases} \quad (2)$$

$$\text{ApplyRopeK}_{l,j}(\boldsymbol{k}) := \begin{cases} \text{ApplyRope}(\boldsymbol{k}, \boldsymbol{p}[j-1]) & \text{if } l > 3 \\ \text{ApplyRope}(\boldsymbol{k}, \boldsymbol{p}[c_{\text{orig}}]) & \text{otherwise,} \end{cases} \quad (3)$$

where $i_{\text{orig}}$ denotes the original position of the given $\boldsymbol{q}$, and $c_{\text{orig}}$ denotes the index of the chunk that the given $\boldsymbol{k}$ comes from, $\boldsymbol{p}_i \in \mathbb{R}^d$ refers to the rotary positional embedding vector for the $i$th position, and the $\text{ApplyRope}(\cdot, \boldsymbol{p}_i)$ function denotes the classic application of RoPE $\boldsymbol{p}_i$ on the given vector as described in Su et al. (2023). The condition $l > 3$ is for applying Relative RoPE instead of Chunk-indexed RoPE; See Appendix D for an in-depth explanation of this choice.

Note that the initial pruning stage of InfiniteHiP's context pruning algorithm runs in $O(T_q T_{kv})$ time, and all subsequent pruning stages run in $O(T_q)$ time. This makes the initial pruning stage the most expensive one as the number of tokens increases. So, asymptotically, InfiniteHiP's context pruning algorithm has a higher time complexity compared to HiP (Lee et al., 2024b). However, since only

two tokens per chunk are accessed and computed at most during the whole process, the SelectRep algorithm can be implemented with a single GPU kernel, without any global synchronizations between each iteration, while providing key sequence dimension parallelism like FlashDecode (Dao et al., 2023) which is not possible in HiP due to internal top-k. This allows InfiniteHiP's context pruning algorithm to run faster in practice with modern GPUs, thanks to its increased parallelism, as shown in Table 4.

Additionally, during decoding, the mask refresh rate of the first pruning stage $n^{(1)}_{\text{refresh}}$ can be set very high without a significant amount of performance degradation, as shown in Table 1. This reduces the impact of the initial pruning stage's latency to the average latency of the entire token generation process.

## A.2 DECODING

In Algorithm 4, we show our decoding algorithm complete with our KV offloading mechanism. In Fig. 7, we visualize the stage caching mechanism in our decoding algorithm.

---

**Algorithm 4** InfiniteHiP Decoding Algorithm

---

**input** The model $\mathcal{M}$, number of layers $L$, number of pruning stages $N$, mask refresh interval $n^i_{\text{refresh}}$.
**output** Generated sequence $y$.
  1: Initialize $y$ with an empty sequence.
  2: $c^{(i)} \leftarrow 0$ for $i = 1 .. N$.
  3: **while** generation has not ended **do**
  4:     **for each** layer $l = 1 .. L$ **do**
  5:         **for each** stage $i = 1 .. N$ **do**
  6:             **if** $(c^{(i)} \bmod n^i_{\text{refresh}}) = 0$ **then**
  7:                 $\mathcal{I}^{(l,i)} \leftarrow$ Run the $i$th pruning stage with $\mathcal{I}^{(l,<i)}$ and the $l$th layer's query and keys with Algorithm 1.
  8:                 Obtain a list of GPU cache misses that occurred during the above process.
  9:             **end if**
10:         **end for**
11:     Perform block sparse attention with $\mathcal{I}^{(l,N)}$.
12:     Obtain a list of GPU cache misses that occurred during the above process.
13:     Evict selected cold tokens from the GPU cache, and replace them with the cache misses, depending on LRU policy.
14:     **end for**
15:     Sample a new token and append it to $y$.
16:     Increment $c^{(i)} \leftarrow (c^{(i)} + 1) \bmod n^{(i)}_{\text{refresh}}$ for $i = 1 .. N$.
17: **end while**

---

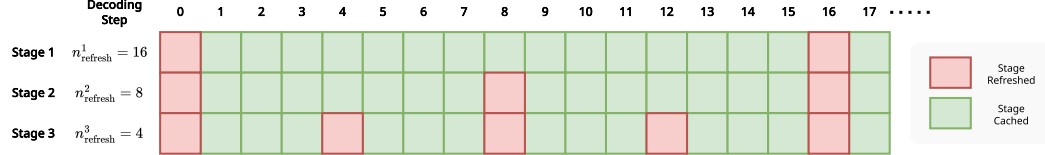

Figure 7: **Visualization of Stage Caching During Decoding.** The visualized mask refresh interval hyperparameter $n^{(1,2,3)}_{\text{refresh}} = (16, 8, 4)$ for simplicity.

## B    VISUALIZATION OF RoPE ADJUSTMENT

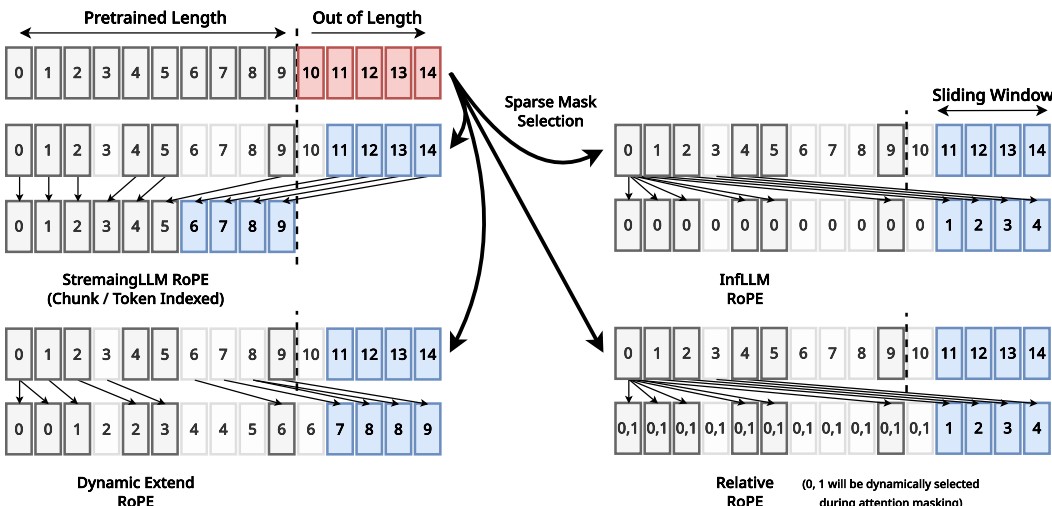

Figure 8: **Visualziation of RoPE Adjustment.**

In Fig. 8, we visualize how we adjust RoPE in more detail. Relative-style RoPE is only used during context pruning because it depends on which branch the token takes during the hierarchical top-1 approximation process. As shown in Table 8, four types of RoPE indexing can be used in masking, and three kinds of RoPE indexing in block sparse attention.

## C    VISUALIZATION OF EACH PRUNING STAGES (MODULES)

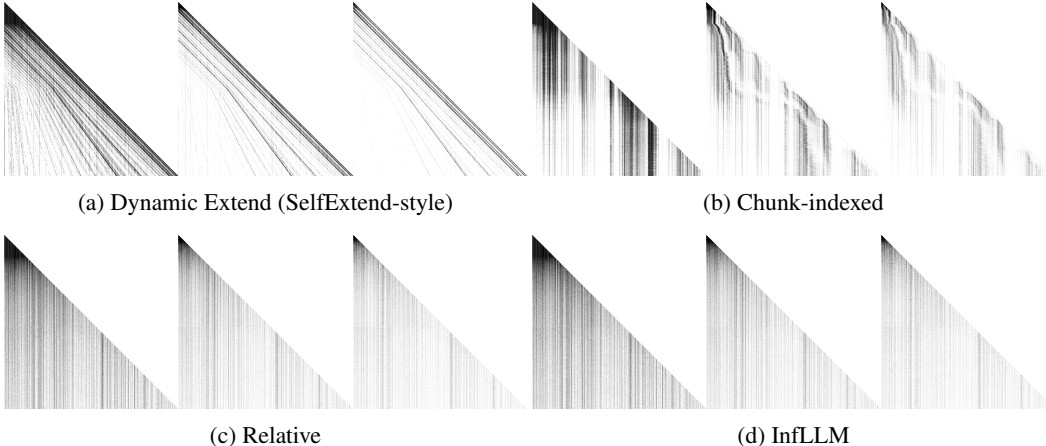

(a) Dynamic Extend (SelfExtend-style)      (b) Chunk-indexed

(c) Relative                                (d) InfLLM

Figure 9: **Visualization of Each Stage of Each RoPE Adjustment Method.** From left, we visualize the output of stages 1, 2, and 3. We use Llama 3.2 1B and T=256K. The model's pretrained context length is 128K. The horizontal axis represents the key sequence dimension, and the vertical axis represents the query sequence dimension. We color non-zero entries in the attention matrix as blocks and masked-out entries as white.

In Fig. 9, we visualize the attention mask generated by various RoPE adjustment methods. In SelfExtend-style RoPE, we extend the RoPE depending on the context length. Therefore, some stretching is observed from the right half of the image beyond the pretrained context length limit. In Chunk-indexed RoPE, we observe curved wiggly artifacts in the second and third stages, which is probably caused by the sliding windows. Since the chunk index position of each token is dynamically

adjusted by previous stages, the sliding patterns change dynamically depending on the inputs. In Relative- and InfLLM-style RoPE, we observe strong vertical patterns because they rely only on the content information in the key vectors rather than the positional information.

## D DISCUSSION ON CHUNK-INDEXED ROPE

This section explains the importance of Chunk-indexed RoPE in addressing the challenges posed by dense attention layers in the baseline HiP model (Lee et al., 2024b). Dense attention layers significantly slow down processing for long-context sequences, particularly when dealing with millions of tokens. While HiP Attention mitigates this issue by limiting experiments to shorter sequence lengths of 32K to 128K due to pretrained context length constraints, our approach must effectively handle much longer sequences, necessitating a more efficient solution.

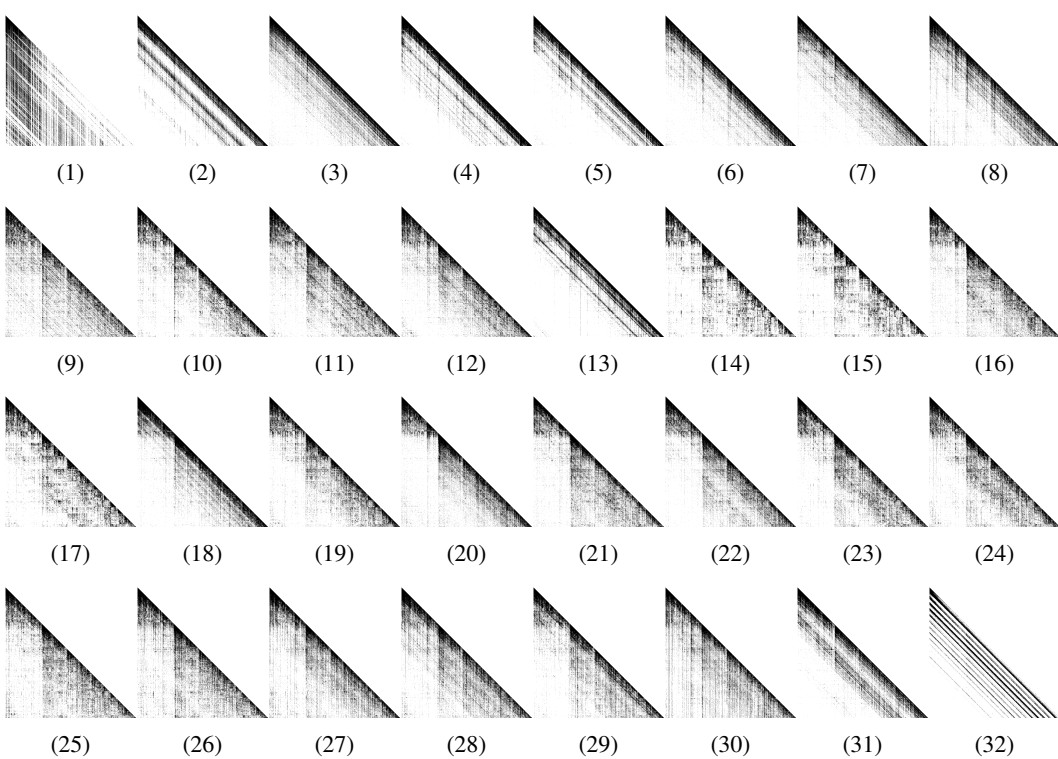

Figure 10: **Generated Mask Example.** We use Llama 3.1 8B with T=64K PG19 sample without the RoPE extend mechanism. Refer Fig. 9 about visualization formats.

Fig. 10 visualizes the attention score patterns. We observe that the earlier layers (e.g., layers up to 5) strongly exhibit dynamic sliding window-like attention, which signifies that these layers focus on relative positional key tokens. This behavior suggests that the model prioritizes positional information in the early layers to establish accurate representations. Once these representations are built, the model can efficiently process long-range information in subsequent layers by leveraging learned semantics instead of positional cues. These observations highlight the critical role of early layers in maintaining coherence while processing large token sequences.

The sliding window patterns in the initial layers play a crucial role in constructing relative positional representations, a task which the block sparse attention struggles to replicate. Block sparse attention often results in staircase-step patterns, leading to inconsistent relative positional attention, as shown in Fig. 11. To address this limitation, we employ two key strategies. First, we increase the retention rate to cover pretrained patterns better by reducing masking jitters. Second, we carefully guide the pruning algorithm using our RoPE adjustment strategies (Chunk-indexed or SelfExtend-style). These adjustments generate sliding window-style artifacts, which leads to sliding window-like masks that

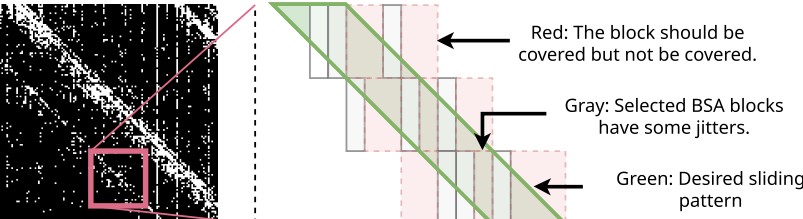

Red: The block should be covered but not be covered.

Gray: Selected BSA blocks have some jitters.

Green: Desired sliding pattern

Figure 11: **Visualization of Approximating Sliding Window with Block Sparse Attention.** (Left) Cropped generated block sparse mask from 13th layer from Fig. 10. White pixels mean non-zero entries in the attention matrix, and black pixels mean masked-out pixels. (Right) Illustration of how sliding windows fail to be approximated by block sparse attention.

effectively capture the diagonal patterns. By integrating these two methods, we minimize the reliance on dense layers while preserving essential positional information.

Table 10: **Performance comparison between relative RoPE only and mixture with Chunk-indexed RoPE.** We use Llama 3.1 8B with context truncation at 300K tokens.

| RoPE style in layers #1–3 | RoPE style in layers #4–32 | InfiniteBench En.MC score (%) |
|---|---|---|
| Relative | Relative | 68.55 |
| Chunk-indexed | Relative | **74.23** |

To validate our configuration, we conduct an empirical ablation study. As shown in Table Table 10, combining Chunk-indexed RoPE and Relative-style RoPE within a single model enhances long-context performance. However, as highlighted in Table Table 8, using Chunk-indexed RoPE in every layer causes significant performance degradation. Thus, our default configuration strategically incorporates both Chunk-indexed and Relative styles, ensuring optimal performance while addressing the efficiency challenges of long-context processing.

# E ADDITIONAL EXPERIMENT RESULTS

## E.1 PASSKEY RESULT ON DEEPSEEK R1 DISTILLED QWEN2

| T (k) | 1000 | 872 | 744 | 616 | 488 | 360 | 232 | 128 |
|---|---|---|---|---|---|---|---|---|
| 100-80% | 100 | 100 | 100 | 100 | 100 | 100 | 100 | 100 |
| 80-60% | 100 | 100 | 100 | 100 | 100 | 100 | 100 | 100 |
| 80-40% | 100 | 100 | 100 | 100 | 100 | 100 | 100 | 100 |
| 40-20% | 100 | 100 | 100 | 100 | 100 | 100 | 100 | 100 |
| 20-0% | 96 | 100 | 100 | 100 | 100 | 100 | 100 | 100 |
| AVG. | 98 | 100 | 100 | 100 | 100 | 100 | 100 | 100 |

Table 11: **Passkey result on DeepSeek R1 Distilled Qwen2 14B.** Each row means the the document location of passkey statement inside of repeated context text. The pretrained context window of Deepseek R1 Distilled Qwen2 is 128K.

In Table 11, we demonstrate our context extension ability on Deepseek R1 Distilled Qwen 2.5 14B (DeepSeek-AI et al., 2025). Our method extends the pretrained context window of Deepseek R1 from 128K to 1M without performance degradation.

## E.2 RULER RESULTS WITH SPARSE DECODE

Table 12: **Average RULER Performance Across Context Length.** The model used is Llama 3.1 8B.

| T (k) | 512 | 256 | 128 | 64 | 32 | 16 | 8 | 4 | Average |
|---|---|---|---|---|---|---|---|---|---|
| FA2 | 0.00 | 0.00 | 74.75 | 86.05 | 89.81 | 93.92 | 94.52 | 96.05 | 66.89 |
| HiP | 0.00 | 0.00 | 26.48 | 63.69 | 84.15 | 93.91 | 94.58 | 95.90 | 57.34 |
| Ours 16K-shallow | 56.92 | 59.90 | 70.25 | 74.60 | 85.22 | 93.55 | 94.79 | 95.89 | 78.89 |
| Ours 5K | 64.68 | 63.74 | 68.21 | 71.62 | 82.33 | 87.89 | 92.09 | 96.41 | 78.37 |
| Ours 3K-5K | 55.77 | 60.05 | 64.73 | 69.39 | 80.83 | 87.75 | 93.41 | 96.26 | 76.02 |
| Ours 3K | 43.98 | 49.50 | 57.63 | 64.68 | 80.10 | 86.16 | 93.24 | 96.70 | 71.50 |

Table 13: **Average RULER Performance Across Subsets.** The model used is Llama 3.1 8B.

| Subset | $NIAH_{SK}^1$ | $NIAH_{SK}^2$ | $NIAH_{SK}^3$ | $NIAH_{MK}^1$ | $NIAH_{MK}^2$ | $NIAH_{MK}^3$ | $NIAH_{MV}$ | $NIAH_{MQ}$ | VR | CWE | FWE | $QA_1$ | $QA_2$ |
|---|---|---|---|---|---|---|---|---|---|---|---|---|---|
| FA2 | 72.5 | 74.0 | 75.0 | 75.0 | 73.5 | 70.0 | 73.4 | 74.4 | 69.9 | 41.5 | 65.1 | 61.5 | 43.8 |
| HiP | 53.0 | 65.8 | 62.8 | 64.3 | 55.4 | 56.8 | 60.0 | 61.8 | 60.7 | 40.6 | 65.5 | 59.4 | 39.5 |
| Ours 16K-shallow | 98.3 | 98.8 | 99.8 | 94.5 | 64.5 | 47.3 | 91.0 | 90.3 | 91.7 | 42.2 | 87.2 | 69.3 | 51.0 |
| Ours 5K | 100.0 | 99.0 | 99.5 | 94.0 | 50.5 | 32.8 | 94.8 | 94.3 | 98.2 | 44.3 | 84.8 | 72.7 | 54.0 |
| Ours 3K-5K | 99.0 | 97.0 | 96.8 | 86.5 | 42.5 | 31.3 | 88.9 | 91.1 | 98.2 | 48.9 | 84.5 | 70.0 | 53.8 |
| Ours 3K | 95.8 | 90.8 | 96.0 | 80.3 | 42.3 | 32.5 | 75.6 | 76.0 | 95.4 | 45.3 | 82.9 | 65.3 | 51.5 |

In Tables 12 and 13, we benchmark the RULER benchmark with InfiniteHiP and baselines in the Llama 3.1 8B model. The baseline (FA2 and HiP) failed to generalize the out-of-length (OOL) situation. 5K and 3K settings are the same as the definition in Appendix F, and 3K+5K uses the 3K setting for prefill and 5K setting for decoding. Lastly, the 16K setting uses a single pruning stage with a chunk size of 32 and uses a 128K sliding window in the first three layers.

We show a near-perfect performance recovery in the RULER benchmark with dense decode in Table 5; however, we could not show good performance recovery in the average score with sparse decode in this section. As shown in Table 13, in most subsets our method shows good performance in most of the sequence length-subset combinations. However, in the multikey 2 and 3 subsets, our method exceptionally fails to find the passkey.

We believe this exceptional failure in the multikey subset is due to distributional failure, rather than retrieval failure. In a multikey subset context, the prefill context is filled with thousands of key-value pairs without priorly knowledge of the target key code. Since the attention mechanism does not know which input tokens are important in prefill tokens, until right before the end-user prompt, the attention probabilities for many input tokens are usually flat and uniform. Therefore, the top-k attention algorithm will select top-k uniformly significant key tokens, which do not recover most of the attention probabilities, compared to a properly concentrated attention matrix row. The more accurately we select the top-k tokens, the more significant the distributional shift (or bias) towards particular key tokens becomes. Therefore, even though we are achieving better performance in natural language understanding and most subsets of RULER, we are failing in this regard. According to concurrent work, Delta Attention (Willette et al., 2025), the distributional shift is the key lacking point of the sparse attention mechanism. By correcting such distributional shifts, we could achieve much better performance even in a multi-key example.

### E.3    ADDITIONAL INFINITEBENCH RESULTS IN VARIOUS MODELS: LLAMA 4, QWEN 2.5, GEMMA2, EXAONE3.5, AND EXAONE3

Table 14: **InfiniteBench Result on Llama 4 Scout 109B.** We replace the NoPE layers with ours, keeping the chunked attention layers. Ours$^{DD}$ means dense attention is used during decoding. FA3 stands for Flash Attention 3 (Shah et al., 2024).

| Method | QA$_{Acc.}$ | QA$_{Recall}$ | MC | Sum | Passkey | Number | KV | Math F. | Code D. | Avg. |
|---|---|---|---|---|---|---|---|---|---|---|
| Ours | 41.83 | 47.71 | 74.67 | 32.62 | 92.71 | 94.58 | 72.40 | 45.43 | 48.48 | 61.16 |
| Ours$^{DD}$ | 43.19 | 48.29 | 74.67 | 34.28 | 100.00 | 99.83 | 99.40 | 41.14 | 49.24 | 65.56 |
| FA3 | 44.34 | 48.82 | 82.10 | 35.27 | 100.00 | 100.00 | 99.20 | 43.14 | 56.85 | 67.75 |

In Table 14, we show the performance of InfiniteHiP with the NoPE-based model, Llama 4 Scout 109 B (AI, 2025). Since the model is using interleaved full attention layers without RoPE per every four layers, the model has natively unlimited context length. Therefore, in Llama 4, we turn off the context extension feature and replace only the full dense layer in every four layers. We could further improve the performance of needle in the haystack by using dense attention during decoding, which is not the dominant latency part in long context question answering.

Table 15: **InfiniteBench Result on Qwen 2.5 32B.** We extend the model context window from 32K up to 384K tokens with ours. FA3 stands for Flash Attention 3 (Shah et al., 2024).

| Method | T (k) | QA$_{Acc.}$ | QA$_{Recall}$ | MC | Sum | Passkey | Number | KV | Math F. | Code D. | Avg. |
|---|---|---|---|---|---|---|---|---|---|---|---|
| Ours | 384 | 27.98 | 44.92 | 75.55 | 27.15 | 98.31 | 100.00 | 67.40 | 36.57 | 40.86 | 57.64 |
| FA3 | 32 | 24.55 | 39.48 | 66.38 | 25.71 | 27.12 | 27.12 | 18.00 | 35.14 | 42.64 | 34.02 |

In Table 15, we show the performance of InfiniteHiP with context extension in Qwen2.5 32B (Qwen et al., 2025). Qwen2.5 is only trained with 32K context, and it can extend the context with YaRN (Peng et al., 2023). Therefore, the trained context window is not long enough to cover all text context of InfiniteBench. With InfiniteHiP, we could improve the benchmark performance from 34.02% to 57.64% (+23.62%) without any further training, while speeding up both prefill and decoding.

In Table 16, we show the performance of InfiniteHiP context extension in Gemma2 (Gemma Team, 2024) and EXAONE (LG AI, 2024a). This table is the raw data of Fig. 4.

### E.4    DETAILED RESULT OF SGLANG END-TO-END DECODING THROUGHPUT

In Tables 17 and 18, we demonstrate the decoding throughput on each system: RTX 4090 24GB and L40S 48GB. This is raw data of Fig. 5. We test only single-batch scenarios because we expect a single sequence to be larger than GPU VRAM. We chose RTX4090 because it is the best consumer-grade GPU and is easily accessible to local LLM end-users; therefore, it will represent real-world decoding

Table 16: **Infinite Bench Results on Gemma2 9B, EXAONE3 and 3.5 7.8B.** OOL (out-of-length) means that the model response is garbage because input context length exceeds model pretrained context length.

| Model | Task | Flash Attention 2 | | | | InfiniteHiP | | | | | | | |
|---|---|---|---|---|---|---|---|---|---|---|---|---|---|
| | | 4 | 8 | 16 | 32 | 4 | 8 | 16 | 32 | 64 | 128 | 192 | 256 |
| | MC (Acc) | 33.62 | OOL | OOL | OOL | 30.57 | 31.00 | 32.75 | 38.43 | 37.12 | 38.43 | 38.86 | 39.30 |
| EXAONE3 | QA (Recall) | 25.80 | OOL | OOL | OOL | 23.12 | 27.57 | 30.03 | 30.77 | 34.85 | 32.83 | 31.89 | 33.41 |
| 7.8B | QA (F1) | 3.92 | OOL | OOL | OOL | 2.84 | 3.63 | 3.93 | 4.66 | 4.95 | 5.52 | 5.30 | 5.04 |
| | Sum (RLsum) | 23.60 | OOL | OOL | OOL | 23.44 | 24.39 | 25.16 | 25.98 | 26.51 | 26.97 | 27.04 | 27.22 |
| | MC (Acc) | 38.43 | 48.91 | 49.34 | 48.91 | 39.30 | 45.41 | 48.91 | 50.66 | 56.33 | 60.26 | 59.39 | 59.83 |
| EXAONE3.5 | QA (Recall) | 20.94 | 27.89 | 31.80 | 40.77 | 19.98 | 24.61 | 30.02 | 35.38 | 41.97 | 46.16 | 47.28 | 47.39 |
| 7.8B | QA (F1) | 8.21 | 10.25 | 11.49 | 11.94 | 8.65 | 10.67 | 13.29 | 15.14 | 16.31 | 17.37 | 18.28 | 17.83 |
| | Sum (RLsum) | 23.00 | 24.48 | 25.97 | 25.81 | 22.66 | 24.00 | 25.22 | 26.23 | 26.67 | 27.08 | 27.12 | 27.17 |
| | MC (Acc) | 42.36 | 48.03 | OOL | OOL | 37.55 | 45.85 | 55.46 | 59.83 | 61.57 | 71.62 | 73.80 | 72.49 |
| Gemma2 | QA (Recall) | - | 22.67 | OOL | OOL | 16.99 | 23.00 | 27.42 | 36.51 | 42.99 | 46.23 | 46.23 | 44.70 |
| 9B | QA (F1) | 11.93 | 12.03 | OOL | OOL | 11.89 | 14.59 | 18.29 | 21.77 | 26.87 | 28.99 | 28.26 | 27.85 |
| | Sum (RLsum) | 20.60 | 21.39 | OOL | OOL | 21.13 | 22.29 | 23.00 | 23.68 | 23.88 | 24.21 | 23.89 | 23.72 |

throughput well. We chose L40S because it is the best cost-performance effective GPU available in Amazon Web Services (AWS) in 2025 to simulate practical serving scenarios.

For the L40S 48GB system, we used the AWS `g6e.48xlarge` node. The specification of the RTX 4090 24GB system is as follows:

| | |
|---|---|
| CPU | AMD Ryzen 7950X, 16 Core, 32 Thread |
| RAM | 128GB, DDR5 5600 Mhz |
| GPU | Nvidia RTX 4090, VRAM 24GB |
| PCIe | Gen 4.0 x8 |
| OS | Ubuntu 22.04.4 LTS |
| GPU Driver | 535.171.04 |

Table 17: **End-to-End Decoding Throughput (token/sec) on RTX4090 24GB.** We use AWQ Llama 3.1 8B with FP8 KV cache data type. We measured the latency of a one batch size with a passkey example. Estimated latencies are measured with estimated attention latency considering previous trends.

| T (k) | 64 | 96 | 128 | 192 | 256 | 384 | 512 | 768 | 1024 |
|---|---|---|---|---|---|---|---|---|---|
| SRT | 88.8 | 74.3 | 63.2 | 49.4 | - | - | - | - | - |
| SRT (Estimated) | 88.8 | 73.8 | 63.2 | 49.0 | 40.1 | 29.3 | 23.1 | 16.3 | 12.5 |
| InfiniteHiP 3K-Fast | 113.3 | 112.5 | 112.0 | 110.6 | - | - | - | - | - |
| InfiniteHiP 3K-Fast (Estimated) | 113.3 | 112.5 | 112.0 | 110.6 | 109.6 | 107.3 | 105.0 | 100.8 | 97.0 |
| InfiniteHiP 3K-Fast (Offload) | 64.5 | 59.6 | 55.9 | 51.1 | 46.6 | 39.9 | 31.8 | 21.6 | 17.3 |
| InfiniteHiP 3K-Flash (Offload) | 66.0 | 62.7 | 60.3 | 58.2 | 56.6 | 53.5 | 49.5 | 44.0 | 40.1 |

Table 18: **End-to-End Decoding Throughput (token/sec) on L40S 48GB.** We use the same setting with Table 17, but the latencies are measured with different GPU, L40S.

| T (k) | 64 | 128 | 256 | 512 | 1024 | 2048 | 3072 |
|---|---|---|---|---|---|---|---|
| SRT | 69.5 | 48.6 | - | - | - | - | - |
| SRT (Estimated) | 69.5 | 48.6 | 30.4 | 17.3 | 9.3 | 4.9 | 3.3 |
| InfiniteHiP | 98.7 | 97.6 | - | - | - | - | - |
| InfiniteHiP 3K-Fast (Estimated) | 98.7 | 97.6 | 95.7 | 92.0 | 85.4 | 74.7 | 66.4 |
| InfiniteHiP 3K-Fast (Offload) | 55.3 | 43.5 | 37.6 | 34.1 | 24.2 | 10.5 | 7.6 |
| InfiniteHiP 3K-Flash (Offload) | 56.6 | 52.0 | 49.4 | 43.7 | 35.2 | 28.0 | 23.8 |

| | | Prefill (ms) | | | | | | | | Decode (us) | | | | | | | |
|---|---|---|---|---|---|---|---|---|---|---|---|---|---|---|---|---|---|
| | T (k) | 32 | 64 | 128 | 256 | 384 | 512 | 768 | 1024 | 32 | 64 | 128 | 256 | 384 | 512 | 768 | 1024 |
| FA2 (1M window) | | 54.6 | 163 | 379 | 821 | 1267 | 1711 | 2602 | 3490 | 213 | 375 | 643 | 1193 | 1787 | 2325 | 3457 | 4645 |
| InfLLM (12K) | | 150 | 178 | 178 | 179 | 180 | 181 | 182 | 183 | 936 | 1145 | 1157 | 1174 | 1167 | 1182 | 1203 | 1222 |
| HiP (1K) | | 68.5 | 95.6 | 109 | 122 | 135 | 135 | 147 | 147 | 330 | 352 | 376 | 399 | 423 | 423 | 446 | 450 |
| **Ours** (3K) | Total | 63.5 | 78.3 | 84.5 | 96.7 | 109 | 122 | 147 | 172 | 81.0 | 83.5 | 89.5 | 103 | 124 | 154 | 195 | 234 |
| | Total (AR) | - | - | - | - | - | - | - | - | 409 | 395 | 425 | 471 | 539 | 559 | 696 | 936 |
| | Stage 0 (%) | 3.3 | 6.2 | 12.6 | 22.9 | 30.8 | 37.2 | 46.7 | 53.7 | 7.4 | 8.4 | 10.5 | 14.7 | 22.5 | 24.9 | 29.5 | 28.2 |
| | Stage 1 (%) | 13.8 | 20.1 | 18.8 | 16.4 | 14.4 | 13.0 | 10.8 | 9.2 | 7.9 | 7.7 | 7.7 | 8.3 | 7.7 | 7.0 | 5.2 | 4.0 |
| | Stage 2 (%) | 33.5 | 28.8 | 26.8 | 23.4 | 20.7 | 18.6 | 15.4 | 13.1 | 11.6 | 11.9 | 11.5 | 10.6 | 9.5 | 8.9 | 7.1 | 5.3 |
| | BSA (%) | 38.9 | 30.7 | 28.4 | 24.8 | 21.9 | 19.7 | 16.4 | 13.9 | 4.0 | 4.0 | 4.6 | 4.8 | 4.0 | 3.7 | 2.9 | 2.2 |
| | Extra (%) | 10.6 | 14.2 | 13.4 | 12.6 | 12.2 | 11.5 | 10.7 | 10.1 | 69.2 | 68.1 | 65.7 | 61.6 | 56.4 | 55.5 | 55.3 | 60.3 |
| **Ours** with Extend (3K) | Total | 103 | 128 | 138 | 158 | 178 | 197 | 236 | 276 | 89.8 | 91.9 | 98.0 | 111 | 133 | 163 | 205 | 245 |
| | Total (AR) | - | - | - | - | - | - | - | - | 425 | 432 | 462 | 520 | 577 | 617 | 842 | 992 |
| | Stage 0 (%) | 3.4 | 6.2 | 12.6 | 22.9 | 31.0 | 37.4 | 47.1 | 53.9 | 6.5 | 7.1 | 9.5 | 16.5 | 22.2 | 26.8 | 28.9 | 31.9 |
| | Stage 1 (%) | 16.9 | 22.1 | 20.6 | 17.9 | 15.8 | 14.3 | 11.9 | 10.3 | 14.5 | 14.7 | 14.2 | 12.6 | 11.2 | 10.5 | 7.8 | 6.7 |
| | Stage 2 (%) | 32.3 | 28.2 | 26.1 | 22.7 | 20.2 | 18.3 | 15.2 | 13.1 | 11.9 | 11.8 | 11.1 | 9.9 | 8.8 | 8.3 | 6.0 | 5.2 |
| | BSA (%) | 44.4 | 34.8 | 32.3 | 28.3 | 25.1 | 22.7 | 18.9 | 16.2 | 4.3 | 4.6 | 5.1 | 5.2 | 4.6 | 4.1 | 2.9 | 2.5 |
| | Extra (%) | 3.0 | 8.6 | 8.4 | 8.2 | 8.0 | 7.3 | 6.9 | 6.5 | 62.8 | 61.7 | 60.1 | 55.9 | 53.3 | 50.3 | 54.4 | 53.7 |

Table 19: **Attention Latency Comparison between InfiniteHiP and Baselines.** Prefill latency is measured with chunked prefill style attention, with a chunk size of 32K. In our rows, *Total* means the average latency of the attention mechanism, *Total (AR)* means the decoding latency without any mask caching mechanism, which is always a mask refreshing scenario, *Stage X* means the latency of X'th pruning stage, *BSA* means the latency of block sparse attention. Ours uses the 3K preset from Table 1.

| | | T=256k | | T=512k | | T=1024k | |
|---|---|---|---|---|---|---|---|
| | | VRAM (GB) | Latency (μs) | VRAM (GB) | Latency (μs) | VRAM (GB) | Latency (μs) |
| FA2 (1M window)* | Runtime | 20.0 (100%) | 1,193 (100%) | 36.0 (100%) | 2,325 (100%) | 68.0 (100%) | 4,645 (100%) |
| InfLLM (12K) | Runtime | 4.8 (23.8%) | 1,186 (99.4%) | 4.8 (13.2%) | 1,194 (51.4%) | 4.8 (6.99%) | 1,234 (26.6%) |
| | Runtime (Fast) | 6.1 (30.4%) | 532 (44.6%) | 6.1 (16.9%) | 902 (38.8%) | 6.1 (8.93%) | 1,864 (40.1%) |
| | Runtime (Flash) | 6.1 (30.4%) | 325 (27.2%) | 6.1 (16.9%) | 475 (20.4%) | 6.1 (8.93%) | 844 (18.2%) |

| | Cached Stages | None | S1 | S1&2 | All | None | S1 | S1&2 | All | None | S1 | S1&2 | All |
|---|---|---|---|---|---|---|---|---|---|---|---|---|---|
| **Ours** with Extend & Offload (3K-fast) | Latency (μs) | 9,803 | 2,579 | 779 | 110 | 19,541 | 4,416 | 836 | 116 | 47,157 | 6,955 | 1,104 | 119 |
| | Stage 0 (μs) | 2,267 | - | - | - | 8,354 | - | - | - | 30,097 | - | - | - |
| | Stage 1 (μs) | 2,854 | 520 | - | - | 3,747 | 1,498 | - | - | 6,192 | 2,903 | - | - |
| | Stage 2 (μs) | 2,247 | 784 | 130 | - | 3,015 | 1,461 | 137 | - | 4,420 | 2,224 | 150 | - |
| | BSA (μs) | 235 | 200 | 37 | 31 | 277 | 177 | 34 | 31 | 326 | 189 | 85 | 30 |
| | Offload (μs) | 2,039 | 869 | 503 | - | 3,901 | 1,110 | 569 | - | 5,857 | 1,533 | 786 | - |
| | Extra (μs) | 161 | 206 | 110 | 79 | 247 | 170 | 96 | 89 | 265 | 106 | 83 | 89 |
| | Mask Hit Ratio (%) | 71.67 | 85.12 | 98.75 | - | 52.66 | 74.74 | 98.42 | - | 28.91 | 56.88 | 98.38 | - |
| | SA Hit Ratio (%) | 58.92 | 69.25 | 88.61 | 99.8 | 54.45 | 68.05 | 89.76 | 99.8 | 51.38 | 67.73 | 88.97 | 99.8 |

Table 20: **Decoding Attention Latency of InfiniteHiP with Offloading.** When *Cached stages* is *None*, all pruning stages from stage 1 through 3 are re-computed, and if it is *All*, then all pruning stages are skipped and only the BSA step is performed. In *S1*, the first stage is skipped, and in *S1&2*, the first two stages are skipped. *Offload* indicates the latency overhead of offloading and the cache management mechanism. The latencies are measured with a single RTX 4090 on PCIe 4.0 x8. The model used is AWQ Llama3.1 with FP8 KV cache. (*) FA2 does not support KV cache offloading and thus cannot run decoding with a context window exceeding 128K tokens using a single RTX 4090. We estimate FA2 results by layer-wise simulation with the same model architecture.

## E.5 DETAILED RESULT OF LATENCY BREAKDOWN

In Tables 19 and 20, we show the detailed latency breakdown numbers of Tables 3 and 4. The latency of each stage is hidden in the main paper, due to space limitations. We can clearly see that stage 0 has a quadratic time complexity, and the other stages have linear complexity. However, Stage 0 is much faster than Flash Attention 2 due to its highly sparse process of quadratic attention matrix computation.

## E.6 DETAILED RESULT OF LONGBENCH

In Table 21, we show the detailed LongBench (Bai et al., 2023) result with InfiniteHiP of Table 2.

| | | Single Document QA | | | Multi Document QA | | | Summarization | | | Few-shot Learning | | | Synthetic | | Code | | Avg. | Avg. |
|---|---|---|---|---|---|---|---|---|---|---|---|---|---|---|---|---|---|---|---|
| | | NQA | Qasper | MFQA | HQA | 2WMQ | MSQ | GR | QMS | MN | TREC | TQA | SAMS | PC | PR | RBP | LCC | Abs. | Rel.(%) |
| Methods | Window | | | | | | | | | | Llama 3 (8B) | | | | | | | | |
| FA2 | 8K | 19.9 | 42.4 | 41.0 | 47.4 | 39.2 | 23.0 | 29.9 | 21.4 | 27.5 | 74.0 | 90.5 | 42.3 | **8.5** | 62.5 | 49.1 | 60.8 | 42.47 | 87.69 |
| Infinite | 8K | 19.4 | 42.8 | 40.4 | 43.8 | 37.9 | 18.3 | 29.3 | 21.4 | **27.6** | 74.0 | 90.1 | 41.7 | 4.5 | 50.0 | 48.6 | 60.1 | 40.62 | 83.23 |
| Streaming | 8K | 20.1 | 42.5 | 39.5 | 43.7 | 37.9 | 19.7 | 29.2 | 21.3 | **27.6** | 73.5 | 90.1 | 41.5 | 5.0 | 49.0 | 49.0 | 60.4 | 40.61 | 83.21 |
| InfLLM | 8K | 22.6 | **43.7** | 49.0 | 49.0 | 35.6 | 26.1 | 30.8 | 22.7 | **27.6** | 73.5 | **90.9** | 42.4 | 7.2 | 84.0 | 46.5 | 59.9 | 44.47 | 92.83 |
| **InfiniteHiP** | **3K** | 26.6 | 43.2 | 50.3 | 51.9 | 41.0 | 30.9 | 31.7 | 23.3 | 26.9 | 75.5 | 90.3 | 43.0 | 7.5 | 93.5 | 64.8 | 63.1 | 47.72 | 100.00 |
| Methods | Window | | | | | | | | | | Mistral 0.2 (7B) | | | | | | | | |
| FA2 | 32K | 22.1 | 29.2 | 47.6 | 37.5 | 22.0 | 19.0 | 31.1 | **23.9** | 26.6 | 71.0 | 86.0 | 42.3 | **4.0** | 86.9 | 54.1 | 57.4 | 41.29 | 96.44 |
| Infinite | 6K | 18.4 | 30.0 | 39.0 | 32.0 | 22.3 | 15.8 | 29.7 | 21.9 | 26.6 | 70.0 | 85.2 | 41.6 | 2.1 | 42.8 | 53.4 | 57.1 | 36.76 | 83.49 |
| Streaming | 6K | 17.9 | **30.1** | 39.1 | 32.2 | 21.8 | 14.7 | 29.8 | 21.9 | 26.6 | 70.0 | 85.6 | 41.3 | 2.5 | 42.2 | 51.5 | 55.4 | 36.41 | 82.63 |
| InfLLM | 6K | 22.1 | 29.3 | 47.4 | 36.6 | 22.3 | 17.7 | 31.0 | 23.5 | **26.7** | 69.0 | 86.7 | 42.5 | 2.9 | 64.0 | 53.0 | 56.7 | 39.46 | 91.23 |
| InfLLM | 12K | 23.0 | 29.5 | 47.6 | 39.5 | **23.6** | 18.9 | 31.4 | 23.8 | **26.7** | 71.0 | 87.3 | 41.8 | 3.0 | **87.4** | 52.1 | 56.7 | 41.46 | 96.99 |
| **InfiniteHiP** | **3K** | 24.1 | 28.7 | **48.6** | 40.4 | 23.2 | **22.1** | 31.6 | 23.8 | 26.5 | 70.5 | **88.8** | 42.7 | 3.5 | 86.6 | 62.1 | 60.4 | **42.71** | 99.85 |

Table 21: **LongBench.** See Table 1 for notations.

| | | Synthetic Tasks | | | | | NLU | | | | Avg. | Avg. |
|---|---|---|---|---|---|---|---|---|---|---|---|---|
| | | RPK | RN | RKV | MF | Avg. | MC | QA | SUM | Avg. | Abs. | Rel.(%) |
| Method | Window | | | | | | Llama 3 (8B) | | | | | |
| FA2 | 8K | 8.50 | 7.80 | 6.20 | 21.70 | 11.05 | 44.10 | 15.50 | 24.70 | 28.10 | 19.57 | 47.83 |
| NTK | 128K | 0.00 | 0.00 | 0.00 | 2.60 | 0.65 | 0.00 | 0.40 | 6.40 | 2.27 | 1.46 | 3.65 |
| SelfExtend | 128K | 100 | 100 | 0.20 | 22.60 | 55.70 | 19.70 | 8.60 | 14.70 | 14.33 | 35.02 | 67.81 |
| Infinite | 8K | 6.80 | 7.60 | 0.20 | 20.60 | 8.80 | 41.50 | 14.60 | 20.80 | 25.63 | 17.22 | 42.52 |
| Streaming | 8K | 8.50 | 8.30 | 0.40 | 21.40 | 9.65 | 40.60 | 14.30 | 20.40 | 25.10 | 17.38 | 42.53 |
| H2O | 8K | 2.50 | 2.40 | 0.00 | 6.00 | 2.73 | 0.00 | 0.70 | 2.80 | 1.17 | 1.95 | 3.95 |
| InfLLM | 8K | 100 | 99.00 | 5.00 | **23.70** | 56.92 | 43.70 | 19.50 | 24.30 | 29.17 | 43.05 | 89.07 |
| **InfiniteHiP** | **3K** | 99.83 | 97.46 | 9.60 | 17.71 | 56.15 | 57.21 | 26.94 | 24.89 | 36.35 | 46.25 | 98.17 |
| **InfiniteHiP** | **3K-fast** | 99.83 | 97.29 | 8.20 | 17.71 | 55.76 | **58.08** | 27.16 | 24.96 | **36.73** | 46.25 | 98.35 |
| **InfiniteHiP** | **3K-flash** | 99.83 | 97.46 | 8.89 | 18.00 | 56.04 | 56.77 | 26.63 | **25.00** | 36.13 | 46.09 | 97.78 |
| **InfiniteHiP** | **5K** | 100 | 99.83 | **10.80** | 20.00 | **57.66** | 55.90 | 30.99 | 22.63 | 36.50 | **47.08** | **99.69** |
| Method | Window | | | | | | Mistral 0.2 (7B) | | | | | |
| FA2 | 32K | 28.80 | 28.80 | 14.80 | 20.60 | 23.25 | 44.50 | 12.90 | 25.90 | 27.77 | 25.51 | 58.37 |
| NTK | 128K | 100 | 86.80 | 19.20 | 26.90 | 58.23 | 40.20 | 16.90 | 20.30 | 25.80 | 42.01 | 77.23 |
| SelfExtend | 128K | 100 | 100 | 15.60 | 19.10 | 58.67 | 42.80 | **17.30** | 18.80 | 26.30 | 42.49 | 78.30 |
| Infinite | 32K | 28.80 | 28.80 | 0.40 | 16.30 | 18.57 | 42.80 | 11.40 | 22.50 | 25.57 | 22.07 | 51.97 |
| Streaming | 32K | 28.80 | 28.50 | 0.20 | 16.90 | 18.60 | 42.40 | 11.50 | 22.10 | 25.33 | 21.97 | 51.62 |
| H2O | 32K | 8.60 | 4.80 | 2.60 | 26.90 | 10.72 | 48.00 | 15.60 | 24.40 | 29.33 | 20.03 | 52.98 |
| InfLLM | 16K | 100 | 96.10 | **96.80** | 25.70 | **79.65** | 43.70 | 15.70 | 25.80 | 28.40 | 54.02 | 94.77 |
| **InfiniteHiP** | **3K** | 100 | 97.97 | 60.80 | 28.00 | 71.69 | 55.46 | 12.74 | **25.86** | 31.35 | 51.52 | 94.44 |
| **InfiniteHiP** | **3K-fast** | 100 | 97.63 | 52.80 | 28.29 | 69.68 | 55.46 | 12.66 | 23.79 | 30.63 | 50.16 | 92.04 |
| **InfiniteHiP** | **5K** | 100 | 99.51 | 83.60 | **29.71** | 78.21 | **56.33** | 14.67 | 24.14 | **31.71** | **54.96** | **99.09** |

Table 22: ∞**Bench Results.** The average score of each category is the mean of dataset performance, and the average score of the whole benchmark is the relative performance compared to the best-performing result. In the 'Window' column, 'fast' and 'flash' indicate refreshing the sparse attention mask less frequently (see Section 4.1). *FA2* refers to truncated FlashAttention2, *Infinite* refers to LM-Infinite, and *Streaming* refers to StreamingLLM. The 'Avg. Rel.' column shows the average of the *relative score* of each subset. The relative score is computed by dividing the original score by the highest score in its column. We believe that the relative score better represents the differences in performance because the variance is normalized per subset.

## E.7 DETAILED RESULT OF INFINITEBENCH WITH EXTRA MODELS

In Table 22, we show the detailed InfiniteBench (Zhang et al., 2024) results with extra models (Jiang et al., 2023).

# F HYPERPARAMETERS

We use the following default setting across our experiments unless stated otherwise:

| | | |
|---|---|---|
| $n_{\text{sink}}$ | Number of sink tokens | 256 |
| $n_{\text{stream}}$ | Number of streaming tokens | 1024 |
| $N$ | Number of pruning stages | 3 |
| $b_q^{(1,2,3)}$ | Query block size (Stage 1, 2, 3) | 64 |
| $l_c^{(1,2,3)}$ | Chunk size (Stage 1, 2, 3) | 256, 32, 8 |
| $k^{(1,2)}$ | Tokens to keep (Stage 1, 2) | 32K, 8K |
| $k^{(3)}$ | Tokens to keep (Stage 3) | *(see below)* |
| $n_{\text{refresh}}^{(1,2,3)}$ | Mask refresh interval (Stage 1, 2, 3) | 16, 8, 4 |

We set $k^{(3)} = 2048$ (4096 for $l \leq 3$) for the default 3K window preset and $k^{(3)} = 4096$ for the 5K window preset. For the 'fast' and 'flash' settings used for the specified rows in Tables 1 and 3, we use $(n_{\text{refresh}}^{(1)}, n_{\text{refresh}}^{(2)}, n_{\text{refresh}}^{(3)}) = (32, 16, 8)$ (fast) and $(96, 24, 8)$ (flash) each, with all other hyperparameters unchanged from the default setting.

We use the following 5K setting across our experiment unless stated otherwise. The unmentioned hyperparameters are the same as with a default setting:

| | | |
|---|---|---|
| $l_c^{(1,2,3)}$ | chunk size (stage 1, 2, 3) | 64, 32, 16 |
| $k^{(1,2)}$ | tokens to keep (stage 1, 2, 3) | 32K, 16K, 4K |

## G    REMAINING CHALLENGES AND FUTURE DIRECTIONS

While our novel framework enhances the speed and memory efficiency of Transformer inference, several challenges yet remain in long-context processing.

First, the issues related to InfiniteHiP are as follows:

- The combination of pruning modules should be studied more in future research. In this study, we focus on introducing a novel sparse attention framework based on a novel modular hierarchical pruned attention mechanism. However, we discovered numerous module design choices during our research. For example, increasing block sizes can reduce latency in masking and increase the retention rates. However, this comes at a cost of performance loss in NLU tasks (e.g., LongBench and InfiniteBench) that require more fine-grained masking. Conversely, larger block sizes can enhance local context retention (e.g., passkey and UUID, which are used in synthetic tasks). These trade-offs highlight the potential for future research into task-dependent module configurations.

Secondly, the issues related to general challenges in serving long-context language models are as follows:

- Significant bottlenecks in the prefill stage. Even after replacing the quadratic attention mechanism with an near-linear alternative like InfiniteHiP, serving over 1M tokens still takes more than 10 minutes in many consumer grade hardwares. While this is significantly faster than Flash Attention 2, it remains impractical for end-users—after all, who would use ChatGPT if it took over 10 minutes just to generate the first token? Thus, reducing or eliminating TTFT (time to first token) and prefilling will be critical for future serving systems. We believe strategies such as lazy initialization and speculative inference—similar to prior work (Fu et al., 2024; Lee et al., 2023) will be essential. Moreover, InfiniteHiP is well-suited for both attention speculation and main forward computation, as it can approximate attention patterns akin to Lee et al. (2024a).

  Despite achieving linear complexity, current Transformer architectures still result in long wait times for users with long-context prompts. While some argue that better hardware and distributed inference will resolve this issue, we see these approaches as neither scalable nor future-proof. Instead, we aim to enhance InfiniteHiP to efficiently handle extremely long contexts while maintaining limited computational costs and achieving significant speedups with practical latencies.

- The linear growth of memory. Although we use KV cache offloading in InfiniteHiP to save GPU memory, in practice we are still limited to CPU memory, which is around 2TB (512GB per GPU; AWS provides around 2TB CPU memory for 8 GPU machines). At this point, we have several options: KV quantization (Hooper et al., 2024), KV eviction (Li et al., 2024; Willette et al., 2024), KV compression (DeepSeek-AI et al., 2024). However, we believe that linear memory complexity is necessary to achieve superior AI models because it enables ability to retain all previously processed information. Therefore, it is crucial to further improve KV cache memory efficiency with quantization and compression. In this regard, our KV cache offloading framework will provide a practical foundation for efficiently managing large working sets.

