# OpenReview forum: "InfiniteHiP: Extending Language Model Context Up to 3 Million Tokens on a Single GPU"
_ICLR.cc/2026/Conference — Submitted to ICLR 2026_

### Official Review · Reviewer_ZK9U · 2025-10-16

**Soundness:** 3
**Presentation:** 3
**Contribution:** 1
**Rating:** 2
**Confidence:** 4

**Summary:**

InfiniteHiP improves the KV cache offloading mechanism of HiP Attention (ICLR 2025) by enhancing its cache management policy. The core idea remains the same to manage the KV cache on the unified memory space while keeping a smaller key bank on the GPU memory, which acts as a cache. The use of the Least Recently Used (LRU) policy as the eviction mechanism is incremental.


After reviewing section 3, FROM HIP TO INFINITEHIP, we are certain that this work is incremental. The token pruning is borrowed from H2O; the dynamic RoPE adjustment is a trick; and Least Recently Used (LRU) is incremental. This is an engineering-heavy paper with incremental improvements over existing work, overstated claims, and limited novel insights. To maintain the high standard of the ICLR conference, we tend to reject this paper.

**Strengths:**

The work integrates sparse attention, offloading, and OOL generalization into one unified system. The training-free design and work integration can lead to better performance.

We believe training-free inference is essential for effective inference, and this paper demonstrates it.

GPU kernels for InfiniteHIP are a good implementation.

**Weaknesses:**

The experimental benchmark selection is LongBench and ∞Bench to prove the effectiveness of InfiniteHiP. However, the context length of LongBench (32K) and ∞Benc (100k) is much lower than its claim of supporting 3 MILLION TOKENS on a single GPU. That means the extended context length has not been proven effective for extremely long context tasks. We suggest that the authors conduct experiments on LongBench v2 with a longer context length.

In Table 5, the RULER Performance of InfiniteHiP starts to be lower than full attention at 128k (74.99 vs. 76.89). Will this tend to continue to go down for a longer context > 128k? This trend can make the title up to 3 million tokens on a single GPU an overstated claim if the InfiniteHiP can not maintain accuracy for long context.

The RoPE Strategy of sing chunk-indexed RoPE for layers 1-3 and relative RoPE for layers 4-32 is based on observing "sliding window patterns in early layers" (Appendix D). Why exactly layers 1-3? What about layers 1-8 or other setting? An ablation study in other settings would help a lot.

The baseline is also out of date, which compares FA2 instead of FA3 [1] or flashinfer [2]. Other lossy baselines include H2O, StreamingLLM, and InfLLM, from 2023-2024. We recommend a state-of-the-art baseline like [3] or [4]

[1] Ye Z, Chen L, Lai R, et al. Flashinfer: Efficient and customizable attention engine for llm inference serving[J]. arXiv preprint arXiv:2501.01005, 2025.

[2] Shah J, Bikshandi G, Zhang Y, et al. Flashattention-3: Fast and accurate attention with asynchrony and low-precision[J]. Advances in Neural Information Processing Systems, 2024, 37: 68658-68685.

[3] Song W, Jayanthi S M, Ronanki S, et al. Compress, Gather, and Recompute: REFORMing Long-Context Processing in Transformers[J]. arXiv preprint arXiv:2506.01215, 2025.

[4] Deng W, Yang Y, Du P, et al. HGCA: Hybrid GPU-CPU Attention for Long Context LLM Inference[J]. arXiv preprint arXiv:2507.03153, 2025.

**Questions:**

Analysis of the impact of InfiniteHIP on network reasoning capabilities？

How would chunk size affect the InfiniteHiP performance?

---

> ### Author Response · Authors · 2025-11-21
> **Response**
>
> Thank you for your time and effort in reviewing our work and recognition of our solid results. We will respond to each of your comments below.
>
> ----
>
> > W1: The experimental benchmark selection is LongBench and ∞Bench to prove the effectiveness of InfiniteHiP. However, the context length of LongBench (32K) and ∞Bench (100k) is much lower than its claim of supporting 3 MILLION TOKENS on a single GPU. That means the extended context length has not been proven effective for extremely long context tasks. We suggest that the authors conduct experiments on LongBench v2 with a longer context length.
>
> Thank you for the suggestion. We have looked at LongBench v2, and we found that it consists of the same benchmarks as ∞Bench. Although LongBench v2 advertises up to 2M token length, its median length is around 100k, the same as ∞Bench. The primary difference between ∞Bench and LongBench v2 resides in the evaluation method, which we do not consider important for evaluating long-context performance. Therefore, we point to our ∞Bench results in Table 1.
>
> > W2: In Table 5, the RULER Performance of InfiniteHiP starts to be lower than full attention at 128k (74.99 vs. 76.89). Will this tend to continue to go down for a longer context > 128k? This trend can make the title up to 3 million tokens on a single GPU an overstated claim if the InfiniteHiP can not maintain accuracy for long context.
>
> Based on the RULER performance shown in Table 5, where InfiniteHiP drops below full attention at 128k tokens (74.99 vs. 76.89), it is a valid concern that this accuracy gap may widen as the context length increases beyond 128k.
>
> While Figure 4 demonstrates that InfiniteHiP can maintain high accuracy when extending short-context models (like Gemma2 and Exaone3.5) up to 250k tokens, we currently lack experimental results to demonstrate InfiniteHiP's accuracy in the 1M to 3M token range.
>
> We acknowledge that, without sufficient evidence to confirm accuracy is maintained at the maximum claimed context, the title, "up to 3 million tokens on a single GPU," is overstated regarding performance. We will tone down our claims in the revised version of the paper to reflect the practical limits of our current accuracy evaluation.
>
> > W3: The RoPE Strategy of sing chunk-indexed RoPE for layers 1-3 and relative RoPE for layers 4-32 is based on observing "sliding window patterns in early layers" (Appendix D). Why exactly layers 1-3? What about layers 1-8 or other setting? An ablation study in other settings would help a lot.
>
> It is noted by earlier works \[1, 2] that attention in earlier layers need "special treatment" than others. Based on this and the attention pattern visualization, we picked layers 1-3 in a heuristic manner.
>
> \[1]: https://arxiv.org/abs/2312.04985
> \[2]: https://arxiv.org/abs/2406.09827
>
> > W4: The baseline is also out of date, which compares FA2 instead of FA3 [1] or flashinfer [2]. Other lossy baselines include H2O, StreamingLLM, and InfLLM, from 2023-2024. We recommend a state-of-the-art baseline like [3] or [4]
>
> We believe that our diverse choice of baselines (not limited to FlashAttention2, LM-Infinite, StreamingLLM, H2O, and InfLLM, but also newer ones such as MInference, FlexPrefill, SeerAttn, and Xattn, shown in Table 5) is enough to position our method within the world of practical efficient attention methods.
>
> Below is an addition comparison of latencies against MInference and FlexPrefill.
>
> **Table: Single-layer Prefill Latency Comparisons (RTX 4090)** *(Unit: ms, best performance in bold, worst in italics)*
> | Length | 32k | 64k | 128k | 256k |
> |-|--|--|--|--|
> |FA2| **34.27** | 119.77 | 462.39 | 1858.60 |
> |Minference| *135.28* | *395.92* | 1083.66 | 2559.47 |
> |FlexPrefill| 100.47 | 350.28 | *1364.37* | *5910.04* |
> |**Ours**| 53.61 | **118.53** | **255.05** | **562.24** |
>
> > Q1: Analysis of the impact of InfiniteHIP on network reasoning capabilities？
>
> In our other research on LLM multi-agent reasoning, we are currently using our InfiniteHiP framework for serving Qwen3 235B, and so far it has delivered good reasoning capabilities. While formal analysis still needs to be done, we have faced no noticeable degradadations in practice.
>
> > Q2: How would chunk size affect the InfiniteHiP performance?
>
> A smaller chunk size helps with the accuracy in benchmarks, but inevitably increases latency. We recommend using the default values (256, 32, 8) for each pruning stage, as shown in Appendix F.
>
> ----
> Thank you again for your evaluation of our work, if you have any remaining concerns please do not hesitate to reach out for the duration of the discussion period.

---

> ### Comment · Reviewer_ZK9U · 2025-11-25
>
> Thank you for your detailed rebuttal.
>
> While I appreciate the authors’ response, I remain concerned about the overstated claims and the limited novelty. As these issues are substantive, I will maintain my original evaluation. I welcome any additional clarification the authors may offer.
>
> Thank you again for your efforts and contribution.

---

### Official Review · Reviewer_G1Ti · 2025-10-21

**Soundness:** 2
**Presentation:** 2
**Contribution:** 3
**Rating:** 4
**Confidence:** 3

**Summary:**

This paper introduces InfiniteHiP, a training-free inference framework designed to address the challenges of processing extremely long contexts in Large Language Models (LLMs). The work tackles three main issues: the high computational and memory costs of the attention mechanism, the failure of pre-trained models to generalize beyond their training length, and the significant GPU memory pressure from the Key-Value (KV) cache. The core contributions are: 1) A modular multi-stage hierarchical token pruning algorithm that dynamically eliminates irrelevant context to accelerate attention. 2) A dynamic RoPE adjustment method that enables out-of-length generalization without fine-tuning. 3) An efficient KV cache offloading system that uses host memory and an LRU policy to manage the cache on a single GPU. The authors demonstrate that InfiniteHiP can process up to 3 million tokens on a single 48GB GPU, achieving significant speedups and strong performance on long-context benchmarks.

**Strengths:**

- The paper is evaluated on a comprehensive set of benchmarks, including LongBench, RULER, and ∞Bench.

- The work is substantial, integrating multiple techniques (sparse attention, OOL generalization, and KV cache offloading) into a single, practical framework. The implementation within the SGLang framework and detailed performance analysis show a significant engineering effort.

- The proposed method achieves strong performance.

**Weaknesses:**

- Crucial details of the proposed method, particularly the complete algorithms for context pruning (Algorithms 1-3), are deferred to the appendix. While this may be due to space constraints, it makes it challenging for the reader to fully grasp the core mechanism without frequently referencing the appendix.

- The heuristic used in the `SelectRep` algorithm is a primary concern. The paper states that when a chunk is divided into two branches, the **first token** of each branch is used as a proxy to decide which branch to discard . This choice seems counter-intuitive. Considering the nature of the causal attention mask, the **last token** of a branch would likely be a more representative summary of the information within that branch. However, even so, the assumption that a single, fixed-position token can reliably represent an entire chunk is not convincingly justified and lacks strong empirical support in the paper.

- The paper could be strengthened by discussing and comparing its KV cache offloading mechanism with other recent works[1,2,3].

I am willing to raise my score if my concerns are adequately addressed.

[1] InfiniGen: Efficient Generative Inference of Large Language Models with Dynamic KV Cache Management

[2] Arkvale: Efficient generative llm inference with recallable key-value eviction

[3] OmniKV: Dynamic context selection for efficient long-context LLMs

**Questions:**

1.  A significant contribution of this work is the sophisticated KV cache management system. Given its practicality, do the authors plan to open-source the code to facilitate reproducibility and encourage further research in this area?

2.  Could the author share insights on why the first token was chosen as the representative token?

---

> ### Author Response · Authors · 2025-11-21
> **Response**
>
> Thank you for your time and effort in reviewing our work and recognition of our solid results. We will respond to each of your comments below.
>
> ----
>
> > W1: Crucial details of the proposed method, particularly the complete algorithms for context pruning (Algorithms 1-3), are deferred to the appendix. While this may be due to space constraints, it makes it challenging for the reader to fully grasp the core mechanism without frequently referencing the appendix.
>
> Thank you for the helpful suggestion. Due to space constraints, we had decided to focus on the novelty of our method in the main text, and defer the details to the appendix. However, we realize that it made the paper harder to follow. Following the reviewer's advice, in the revision, we will try to integrate more of the details in the main text.
>
> > W2: The heuristic used in the SelectRep algorithm is a primary concern. The paper states that when a chunk is divided into two branches, the first token of each branch is used as a proxy to decide which branch to discard. This choice seems counter-intuitive.
>
> There was a mistake in our writing of Appendix A; We do not select the first token, but rather the *center* token of each branch. We apologize for the error. We have changed our implementation of SelectRep multiple times, and it seems we forgot to update the writing accordingly.
>
> > However, even so, the assumption that a single, fixed-position token can reliably represent an entire chunk is not convincingly justified and lacks strong empirical support in the paper.
>
> This issue was explored extensively in the original HiP Attention paper (see Section 4 and Appendix A of \[1]), which is the reason we decided not to include it in our paper. To summarize their findings: Since key vectors close together in time exhibit higher similarity in attention scores, the center token, which is closer to the other tokens in the branch than any other token, is the best one for the representative of the branch (over all other tokens).
>
> \[1]: https://arxiv.org/abs/2406.09827
>
> > W3: The paper could be strengthened by discussing and comparing its KV cache offloading mechanism with other recent works[1,2,3].
>
> Thank you for the suggestion. InfiniGen uses the singular value decomposition (SVD) to generate partial weights, which is used to approximate the attention matrix ahead of time, which is used to load the actual KV values. InfiniteHiP instead uses the sparse mask obtained during the mask estimation algorithm, which allows us to avoid the expensive SVD pre-processing. The results of each pruning stage is masked for a certain number of steps, which further reduces the KV loading cost.
>
> > Q1: A significant contribution of this work is the sophisticated KV cache management system. Given its practicality, do the authors plan to open-source the code to facilitate reproducibility and encourage further research in this area?
>
> We have already open-sourced the code for InfiniteHiP, since we share the same ideas as the reviewer. However, due to the double-blind review policy, we cannot disclose the address here.
>
> > Q2: Could the author share insights on why the first token was chosen as the representative token?
>
> (See W2)
>
> ----
> Thank you again for your evaluation of our work, if you have any remaining concerns please do not hesitate to reach out for the duration of the discussion period.

---

> > ### Comment · Reviewer_G1Ti · 2025-11-24
> >
> > Thank you for your rebuttal. Could you please update the content in the paper's PDF? Additionally, I noticed that InfBench and LongBench used Llama-3-8B, while Ruler used Llama-3.1-8B. Could you please also evaluate InfBench and LongBench using Llama-3.1-8B? This would facilitate the performance comparison between the Full Attention baseline (FA2, 128k window len) and InfiniteHiP. Furthermore, the related work section still seems a bit weak; please elaborate on the similarities and differences between methods to clearly establish your contributions.
> >
> > These are my main concerns, and I will increase my score if they are addressed.

---

### Official Review · Reviewer_SD7R · 2025-10-25

**Soundness:** 3
**Presentation:** 2
**Contribution:** 3
**Rating:** 6
**Confidence:** 2

**Summary:**

InfiniteHiP is a training-free long-context inference framework designed to address three key bottlenecks of LLMs when processing long sequences: computational efficiency, memory consumption, and generalization beyond the pretraining window.
Building upon the original HiP, InfiniteHiP introduces a series of system-level improvements that make long-context inference feasible on a single GPU. The framework consists of three major components: Hierarchical Multi-Stage Pruning; Dynamic RoPE Adjustment, which adapts positional encoding strategies dynamically to enable out-of-length generalization for short-context pretrained models; and Hierarchical KV Offloading with LRU Policy, which manages multi-stage cache refreshing and memory transfer between GPU and host to minimize VRAM pressure. Through the synergy of these mechanisms, InfiniteHiP achieves significant performance improvements within the SGLang inference framework, specifically, a 7.24× end-to-end decoding speedup and an 18.95× acceleration in attention computation on million-token contexts, all without requiring any retraining.

**Strengths:**

1. The work demonstrates strong practicality and engineering significance. InfiniteHiP can be directly integrated with a variety of existing models, such as LLaMA, Qwen, Gemma, and EXAONE, providing a general and deployment-ready solution for long-context inference on commodity GPUs.
2. Another notable strength lies in its unified and system-oriented design perspective. Instead of focusing on a single optimization aspect, the framework simultaneously tackles the three major challenges of long-context modeling: computation, generalization, and memory through a coherent modular architecture.

**Weaknesses:**

1. Despite its strong engineering impact, the scope of related work is relatively limited, covering only four prior studies, which may not sufficiently position InfiniteHiP within the broader literature of efficient attention and memory optimization.
2. The main innovations reside at the system level, and the algorithmic novelty is incremental rather than conceptual. Each of the three modules, pruning, RoPE adjustment, and KV management, builds upon previously established ideas, leading to the impression of being “incremental but practical.”
3. Although several ablation experiments are presented, the paper lacks a systematic quantitative analysis that isolates and justifies the independent contribution of each module. Strengthening the analytical rigor and theoretical interpretation of these components would significantly enhance the paper’s scientific depth and persuasive power.

**Questions:**

see weakness

---

> ### Author Response · Authors · 2025-11-21
> **Response**
>
> Thank you for your time and effort in reviewing our work and recognition of our solid results. We will respond to each of your comments below.
>
> ----
>
> > W1: the scope of related work is relatively limited, covering only four prior studies.
>
> We believe that our diverse choice of baselines (not limited to FlashAttention2, LM-Infinite, StreamingLLM, H2O, and InfLLM, but also MInference, FlexPrefill, SeerAttn, and Xattn, shown in Table 5) is enough to position our method within the world of practical efficient attention methods.
>
> > W2: The main innovations reside at the system level, and the algorithmic novelty is incremental rather than conceptual. Each of the three modules, pruning, RoPE adjustment, and KV management, builds upon previously established ideas, leading to the impression of being “incremental but practical.”
>
> Our method is far from incremental; the idea of structuring the attention pruning into multiple modular stages for efficiecy (Section 3.1), is unique to our method. This is a novel conceptual idea that we came up by ourselves, inspired by big corporations' multi-stage interview processes in real life, not a previously established idea in the efficient attention field.
>
> Even if one chooses to ignore the above point, it is not easy at all to combine all the ideas into one practical framework that performs as well as InfiniteHiP. Our paper's contribution lies in the frontier we have pushed in the efficient attention space.
>
> > W3: Although several ablation experiments are presented, the paper lacks a systematic quantitative analysis that isolates and justifies the independent contribution of each module. Strengthening the analytical rigor and theoretical interpretation of these components would significantly enhance the paper’s scientific depth and persuasive power.
>
> We have an extensive analysis section in Section 4.3, which compares throughput with our offloading component off and on, ablation on the number of pruning stages in Table 7, ablation on RoPE interpolation strategies in Tables 8 and 9. In addition, please see Tables 19 and 20 in the appendix, where we break down the latency of each individual component of our method.
>
> ----
> Thank you again for your evaluation of our work, if you have any remaining concerns please do not hesitate to reach out for the duration of the discussion period.

---

### Meta-Review · Area_Chair_bcwQ · 2026-01-04

**Summary:**

This submission presents InfiniteHiP, a training-free inference framework that targets practical extremely long-context serving by combining (i) hierarchical multi-stage token pruning, (ii) dynamic RoPE adjustment for out-of-length generalization, and (iii) hierarchical KV cache offloading with an LRU policy within SGLang. The reviewers agree the work is engineering-intensive and practically relevant。
During rebuttal, the authors clarified important technical details (notably correcting the SelectRep description to use a center token rather than the first token, with justification grounded in the original HiP analysis) and pointed to existing ablations/latency breakdowns. However, key concerns remain.
One reviewer (ZK9U) maintains a strong rejection primarily due to limited novelty and overstated claims regarding “up to 3M tokens”, noting that accuracy is only validated on benchmarks whose lengths are far below the claimed maximum and that a performance gap vs full attention emerges at 128k on RULER. The authors acknowledged the lack of accuracy evidence in the 1M–3M range and indicated they will tone down claims in the revision, which helps but also weakens the headline contribution unless longer-context accuracy is validated. Another reviewer requests updates to the PDF, stronger related-work positioning (especially on KV offloading), and aligned evaluations across model variants (e.g., Llama-3.1-8B across benchmarks) to support fair comparisons and clearer attribution of contributions.

**Reviewer Concerns:**

1. Novelty concerns; the work is incremental (ZK9U, SD7R).

2. The claim of supporting 3M context lacks effective experimental evidence (ZK9U); the paper is evaluated only on LongBench 32k and InfiniteBench 100k, which does not match the 3M claim.

3. Missing baselines and related work (G1Ti).

4. Lack of ablation experiments on the choice of RoPE layers (ZK9U).

**Reviewer Scores:**

Reviewer G1Ti and ZK9U: Very likely to keep their scores unchanged, or make only minor adjustments. The core concerns have not been resolved, including over-claiming, novelty, and the lack of added related work and baselines.

Reviewer SD7R: Likely to maintain the current score, considering that most concerns have been addressed.

---

### Decision · Program_Chairs · 2026-01-26

Reject